# Uncovering Neural Scaling Laws
# in Molecular Representation Learning

**Dingshuo Chen**[1][*]  **Yanqiao Zhu**[2][*]  **Jieyu Zhang**[3]  **Yuanqi Du**[4]
**Zhixun Li**[5]  **Qiang Liu**[1]  **Shu Wu**[1][†]  **Liang Wang**[1]
[1]Center for Research on Intelligent Perception and Computing
Institute of Automation, Chinese Academy of Sciences
[2]Department of Computer Science, University of California, Los Angeles
[3]The Paul G. Allen School of Computer Science and Engineering, University of Washington
[4]Department of Computer Science, Cornell University
[5]Department of Systems Engineering and Engineering Management
The Chinese University of Hong Kong
✉ Primary contact: `dingshuo.chen@cripac.ia.ac.cn`

## Abstract

Molecular Representation Learning (MRL) has emerged as a powerful tool for
drug and materials discovery in a variety of tasks such as virtual screening and
inverse design. While there has been a surge of interest in advancing model-
centric techniques, the influence of both data quantity and quality on molecular
representations is not yet clearly understood within this field. In this paper, we delve
into the neural scaling behaviors of MRL from a data-centric viewpoint, examining
four key dimensions: (1) data modalities, (2) dataset splitting, (3) the role of
pre-training, and (4) model capacity. Our empirical studies confirm a consistent
power-law relationship between data volume and MRL performance across these
dimensions. Additionally, through detailed analysis, we identify potential avenues
for improving learning efficiency. To challenge these scaling laws, we adapt seven
popular data pruning strategies to molecular data and benchmark their performance.
Our findings underline the importance of data-centric MRL and highlight possible
directions for future research.

## 1 Introduction

The research enthusiasm for Molecular Representation Learning (MRL) is steadily increasing,
attributed to its potential in expediting drug and materials discovery processes compared with
conventional *in vitro* and *in vivo* experiments [1–4]. Within the context of MRL, the central objective
is to leverage specific featurizations, or modalities, of molecules in order to learn continuous vector
representations. These representations aim to capture comprehensive chemical semantics and exhibit
high expressiveness, thereby effectively addressing various downstream tasks [5–17].

A trend in the field is developing neural architectures and training strategies to improve the ex-
pressiveness of the learned representations. However, the influence of varying data scales on the
performance of MRL under different learning scenarios is yet to be fully understood. To fill this gap,
we draw attention to the following questions: *What are the neural scaling behaviors of molecular
representation learning? Do they align with the previous scaling laws such as the power-law, es-
tablished in other domains [18–20]?* Beyond common research objects in neural scaling laws such
as the impact of pre-training and model parameter sizes [21, 18, 22], MRL further presents unique

---

[*]These authors made equal contribution to this research.
[†]Corresponding author: Shu Wu (`shu.wu@nlpr.ia.ac.cn`).

37th Conference on Neural Information Processing Systems (NeurIPS 2023) Track on Datasets and Benchmarks.

data-oriented challenges. These include the selection of appropriate modality [23] and issues related to Out-Of-Distribution (OOD) generalization [24].

To provide a comprehensive understanding of these complexities, we investigate the impact of various design dimensions on MRL from a data-centric perspective. Specifically, our exploration spans four core design dimensions: (1) data modalities (molecular featurizations), (2) data splitting, (3) the role of pre-training, and (4) model capacity. We identify five scientific questions and outline our key observations as follows. A succinct summary of contribution of this work is present in Figure 1.

(Section 3.1) **How does performance scale with data quantities?** We conduct extensive experiments on four large-scale molecular property prediction datasets. These datasets contain a number of classification and regression tasks, both in single-task and multi-task settings, focusing on properties ranging from quantum mechanical properties to macroscopic influence on human body. The experimental results indicate that the model performance generally follows a power-law relationship with data quantities. Compared with the neural scaling laws in Natural Language Processing (NLP) and Computer Vision (CV) domains, there is no apparent performance plateau [21] within the range of datasets we explored, regardless of low-data and high-data regimes, which implies that the performance improvement with increasing dataset size in MRL is highly predictable.

(Section 3.2) **How do different molecular modalities influence scaling laws?** Since distinct molecular featurizations might carry different semantic meanings and their corresponding neural encoders have different inductive bias, the selection of appropriate modalities (molecular featurizations) in MRL remains an open question. In our investigation, we specifically choose three commonly used modalities (2D graphs [6, 8, 24], SMILES strings [25–28], and Morgan fingerprints [23]) and compare their performance on three classification tasks. We find that different modalities exhibit distinct learning behaviors in MRL; graphs and fingerprints are identified as the most data-efficient modalities, exhibiting the largest power-law exponent. In comparison, SMILES strings demonstrate lower performance improvement with the same data increment.

(Section 3.3) **Does pre-training consistently result in positive transfer across all data scales in MRL?** While previous studies tend to argue that molecular pre-training can improve performance on downstream tasks, these conclusions are often drawn based on evaluations on the entirety of available datasets [29–33, 30, 34]. Our work further probes into the effects of graph-based pre-training across varying scales of downstream datasets. The results show that pre-training indeed brings improvements in low-data regimes. However, the power-law exponent of the performance curve, reflecting the rate of growth with incremental data sizes, is smaller with pre-training compared to training from scratch. We suppose that pre-training only delivers stable gains when the downstream dataset is relatively small—under 40K samples, for instance. As the downstream dataset scales up, this positive gains diminish and might even revert to a negative transfer in the high-data regime.

(Section 3.4) **What influence do dataset splits exert on scaling laws?** Out-of-Distribution (OOD) generalization poses a great challenge in MRL [24]. Given that the training distributions, which typically encompass known compounds, often differ from the test distributions containing unknown compounds, this divergence is particularly prevalent in drug discovery. However, the impact of dataset splitting strategies on neural scaling laws remains largely unknown. To bridge this gap, we study three splitting strategies: random, imbalanced, and scaffold splitting. These strategies differ primarily in the degree of overlap between their respective training and test sets. Our empirical findings indicate that while all three splitting schemes adhere to the scaling laws, the exponent associated with random splitting is the smallest, suggesting that as the divergence between training and test distributions increases, the efficiency with which the model utilizes data samples decreases. Such results accentuate the inherent challenges of MRL in practical scenarios, which face with ubiquitous problems of distribution shifts.

(Section 3.5) **How does the model capacity affect the scaling laws?** The model capacity stands as another crucial factor impacting the performance of MRL models [18, 22] as it significantly influences the requisite amount of training data. To study the effect of model capacity in MRL, we vary the parameter size of Graph Isomorphism Networks (GIN) [5], a widely adopted graph neural network architecture and examine its performance on three classification tasks. We observe that, while model capacity does affect data utilization efficiency, there is not a clear correlation between the dataset size and the optimal model capacity required to obtain saturation performance. Unexpectedly, we find that the optimal model capacity for a task with a smaller training set could be larger than that of a task with a larger training set, which contradicts our intuition that larger models usually need larger

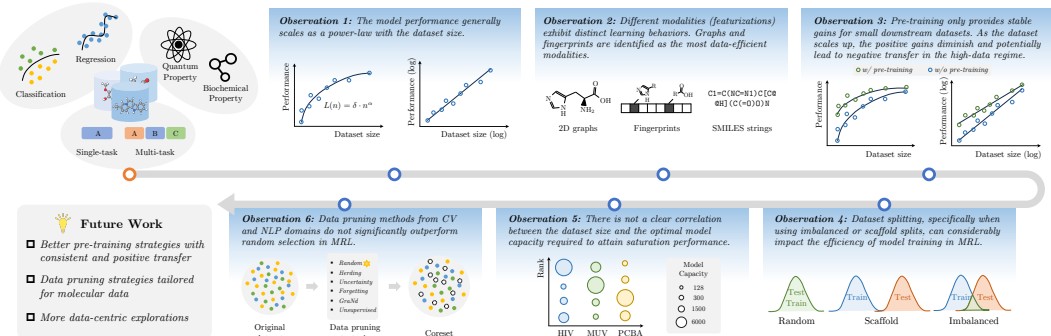

Figure 1: Summary of contributions of this work. We conduct a comprehensive **data-centric** study to examine the neural scaling laws in molecular representation learning. We explore four dimensions affecting the data utilization efficiency: (1) data modalities, (2) the role of pre-training, (3) dataset splitting, and (4) model capacity. Additionally, we study the subset selection problem by adapting seven data pruning strategies to molecular graphs.

training datasets. It suggests the need for further exploration of the interplay between model capacity and data scale, rather than applying a one-size-fits-all approach.

(Section 3.6) **Can a curated subset from the full dataset yield comparable or even superior results?** In CV and NLP domains, the utility of data pruning has been extensively explored due to the computational burden imposed by increasingly large models and massive amounts of data [35–37]. In MRL, however, the extend of data redundancy and the potential of pruning strategies to alleviate computational burden remain largely unexplored. To address this gap, we benchmark seven data pruning strategies originally proposed for image data and adapt them to molecular graph models on three classification tasks. The results show that these data pruning methods do not significantly outperform random selection, which highlights the need for developing data pruning strategies specifically tailored to molecular data.

Based on our empirical analysis, we identify several factors that can enhance data utilizaiton efficiency of MRL. These include the utilization of graph and fingerprint modalities as input data, the application of pre-trained models on small-scale downstream datasets, and the design of data pruning strategies specifically tailored to molecular data. To the best of our knowledge, our study is the first to approach MRL from a data-centric perspective. We envision that this work could provide valuable insights that facilitate future explorations in the field.

## 2 Experimental Settings

In order to answer the six scientific questions through empirical investigation, we conduct a series of experiments on the neural scaling behaviors between model performance and varied data quantities. Specifically, we divide the whole dataset into nine proportional subsets: [1%, 5%, 10%, 20%, 30%, 40%, 60%, 80%, 100%], and for each configuration, we randomly select five seeds and report the mean performance. Then, we study how each design dimension of interest in MRL models influences the scaling laws. To demonstrate the trend of neural scaling laws, we employ the least square model to fit the performance curve.

### 2.1 Datasets and Tasks

Given the potential issues of over-fitting and spurious correlations that may arise with limited samples, we focus on relatively large-scale datasets containing more than 40K molecules in MoleculeNet [38]. Also, these datasets should cover both classification and regression tasks, are diverse in task settings (i.e. single- and multi-task settings), and focus on important biophysics and quantum mechanics properties. As a result, we choose four datasets ranging from molecular-level properties to macroscopic influences on human body for experimental investigation: HIV [39], MUV [40], PCBA [41] and QM9 [42]. Readers can refer to Appendix B for a more detailed discussion on datasets and tasks.

## 2.2 Implementation Details

In the following, we briefly introduce the implementation details of our experiments. Note that our experiments cover multiple dimensions, and thus all experimental settings will remain consistent except for the design dimension to be studied. Please refer to Appendix A for a detailed introduction of implementation details.

**Modalities.** Molecular featurizations translate chemical information of molecules into representations that can be understood by machine learning algorithms. Each featurization can thus be regarded as a modality of the molecular data. Here, we consider the following four molecular modalities: (1) **2D topology graphs** model atoms and bonds as nodes and edges respectively, (2) **3D geometric graphs** include coordinates of atoms into their representation to depict how atoms are positioned relative to each other in 3D space, (3) **Morgan fingerprints** [23] encode molecules into fixed-length bit vectors which map certain structures of the molecule within certain radius of molecular bonds, and (4) **SMILES strings** [25] represent chemical structures in a linear notation using ASCII characters, with explicit information about atoms, bonds, rings, connectivity, aromaticity, and stereochemistry.

**MRL models.** Since our experiments involve four different data modalities, each modality is modeled with its corresponding encoders.

- For 2D graphs, we utilize the Graph Isomorphism Network (GIN) [5] as the encoder. To ensure the generalizability of our research findings, we adopt the commonly recognized experimental settings proposed by Hu et al. [24], with 5 layers, 300 hidden units in each of layer, and 50% dropout ratio.

- For 3D geometries, we employ the widely-used SchNet model [43] as the encoder. We set the hidden dimension and the number of filters in continuous-filter convolution to 128. The interatomic distances are measured with 50 radial basis functions, and we stack 6 interaction layers in SchNet.

- For fingerprints, we first use RDKit [44] to generate 1024-bit molecular fingerprints with a radius $R = 2$, which is roughly equivalent to the ECFP4 scheme [7]. Given the lack of established encoders for fingerprints, we conduct a comparison between two classic encoders, a single-layer MLP and a single-layer Transformer. Please refer to Appendix C.4 for the detailed results. According to the experiments, we choose the Transformer model [45] with 8 attention heads for modeling fingerprints. The bit embedding dimension is set to 64, and the hidden dimension is set to 300.

- For SMILES strings, we employ the same model architecture as the fingerprints to ensure a fair comparison. The only difference is the dictionary size, which will be discussed in Section 3.2.

**Training details.** We follow the settings proposed by Hu et al. [24]. The dataset is split randomly. For classification tasks, we employ an 80%/10%/10% partition for the training, validation, and test sets, respectively. Meanwhile, for regression tasks, the dataset is split into 110K molecules for training, 10K for validation, and another 10K for testing. All models are initialized using the Glorot initialization [46]. The Adam optimizer [47] is employed for training with a batch size of 256. For classification tasks, the learning rate is set at 0.001 and we opt against using a scheduler. For regression tasks, we align with the original experimental settings of SchNet, setting the learning rate to $5 \times 10^{-4}$ and incorporating a cosine annealing scheduler.

**Evaluation metrics.** For `HIV` and `MUV` datasets, performance is measured using the Area Under the ROC-Curve (ROC-AUC), while report the performance on `PCBA` in terms of Average Precision (AP) —higher values in both metrics indicate better performance. When assessing quantum property predictions, the Mean Absolute Error (MAE) is used as the performance metric, with lower values indicating better accuracy.

## 3 Empirical Studies

In this section, we summarize our empirical studies on the aforementioned dimensions of MRL. Firstly, we show the neural scaling laws between model performance and data quantities. Then, we demonstrate the impact of four design dimensions of MRL: data modalities, the role of pretraining, data splits, and model capacity. Lastly, we study data pruning strategies for molecular graphs.

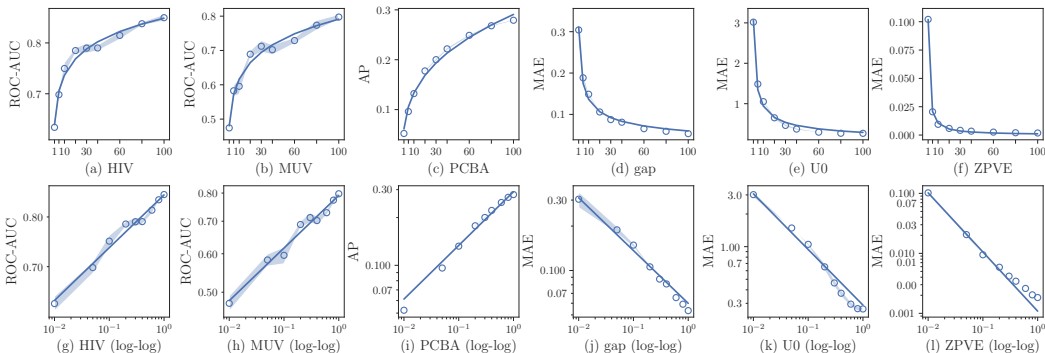

Figure 2: The general neural scaling laws of molecular representation learning. The first row displays the relationship between model performance with respect to varied data sizes in linear coordinates and the second row shows the same on a logarithmic scale.

## 3.1 General Neural Scaling Laws for Molecular Data

Early studies of both classical learning theory and neural scaling laws [48, 18] show that the test performance $L(n)$ increases polynomially with the training data size $n$:

$$L(n) = \delta \cdot n^{\alpha}, \tag{1}$$

where $\delta$ is the coefficient and $\alpha$ is the exponent of the power law. We start our investigation on whether MRL also adheres to this power law relationship with the most common setting of graph-based MRL. Specifically, on the QM9 dataset, we study GINs on 2D graphs for classification tasks and SchNet on 3D geometric graphs for regression tasks. Figure 2 demonstrates the relationship of model performance with respect to the data size across all datasets, where the first row displays the curves in linear coordinates and the second row shows the same on a logarithmic scale.

**Observation 1.** It can be observed that figures in the second row generally exhibit a linear relationship, which suggests alignment with the aforementioned power law. Unlike previous findings in the NLP and CV domains [18], there is no apparent performance plateau within the range of datasets we explored. This pattern remains consistent in both low- and high-data regimes, which suggests that the performance of supervised MRL is highly predictable and consistently improves with increasing data quantity. We note that we observe the same phenomenon consistently when applying other modalities, as we will show later in Figure 3. Furthermore, we also validate the neural scaling laws by examining them on a large-scale dataset PCQM4Mv2 [49] from Open Graph Benchmark (OGB) [50], as well as by employing two advanced 3D GNN encoders PaiNN [51] and SphereNet [52] on QM9. Readers of interest may refer to Appendix C.2 for detailed results and analysis.

## 3.2 Scaling Laws with Different Molecular Modalities

The choice of the most appropriate modalities in MRL remains a topic of continuous discussion. Also, there is a lack of fair comparisons regarding data efficiency across these modalities. In this section, we compare the learning behaviors of the three modalities: 2D graphs, SMILES strings, and fingerprints. To ensure a fair comparison, we maintain roughly equivalent parameter sizes across the modality encoders. Such configurations are selected to maximize the expressiveness of each model. Specifically, we employ a 5-layer GIN model for the 2D graph modality and a 1-layer transformer model for both SMILES strings and fingerprints. It is noting that transformers used for SMILES and fingerprints differ in vocabulary sizes: while the former has a size of 2 for fingerprints due to the binary nature of fingerprint strings, the latter adopts a more expansive vocabulary size of 7,924 for SMILES, following ChemBERTa [26]. We present the results illustrating model performance in relation to different data sizes for the three encoders in Figure 3, with anomalous performance explicitly marked in red.

**Observation 2.** In general, we observe that different modalities exhibit distinct learning behaviors. Among the three modalities, the graph modality consistently demonstrates superior data utilization efficiency across all three tasks, emerging as the most efficient modality for MRL. Fingerprints, characterized with a lower exponent, deliver a slightly lower performance improvement with equivalent

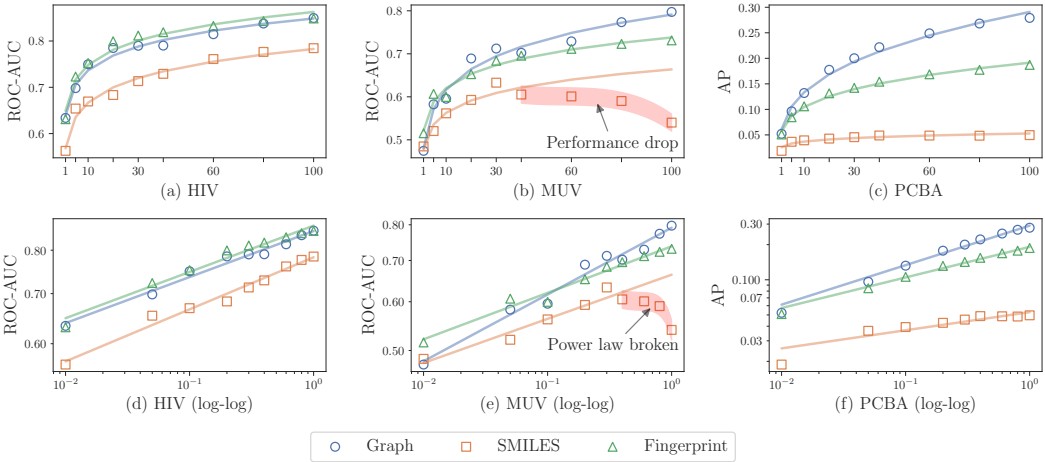

Figure 3: Neural scaling laws with three encoders: 2D graphs, SMILES strings, and fingerprints. Anomalous performance is explicitly marked in red.

data increments. SMILES strings, on the contrary, are the least data efficient modalities. On the MUV dataset, we even notice counter-intuitive performance degradation that breaks the power law [3]. However, it is important to recognize that several studies have underscored the efficacy of language models pre-trained on large-scale SMILES string datasets, which exhibit remarkable performance in the downstream tasks [28, 26]. This points towards the superior transferability of the SMILES-based modality and we will leave this performance gap between pre-trained and those trained from scratch for future research.

## 3.3 Influence of Pre-training on the Scaling Laws

Molecular pre-training studies how to leverage unlabeled molecular data to improve the learned representations [53]. Previous studies have generally suggested that pre-training can induce a positive transfer to downstream tasks [33, 30, 29]. However, we argue that the existing evaluations rely on the entirety of available datasets. We hypothesize that when fine-tuning with datasets of varied sizes, the extent of this positive transfer might fluctuate. Therefore, in this section, we explore whether pre-training consistently leads to positive transfer with different sizes of fine-tuning datasets.

In the experiments, we focus on the state-of-the-art molecular pre-training model GraphMAE [33], which involves masking the atom type of partial atoms in the molecules, re-masking the encoded atom representations from the backbone model, and eventually using a decoder model to reconstruct the original atom features. The GraphMAE model is first pre-trained on PCQM4Mv2 and then fine-tuned on three classification tasks. The results in Figure 4 compares the performance curve with and without pre-training on downstream datasets of varied sizes.

**Observation 3.** Our empirical results reveal that, while pre-trained models still adhere to a power law as their downstream data size varies, they differ from models trained from scratch by having a higher intercept and a lower exponent in the logarithm-scaled plots. This implies that while pre-trained models benefit from better initializations, they demonstrate reduced data efficiency when fine-tuned on downstream datasets.

Our findings also indicate that the benefits of pre-training are particularly pronounced when fine-tuning with smaller datasets and such advantages start to diminish as the size of downstream datasets increases. This is evidenced by the PCBA dataset, where we observe a crossover in performance at a data scale of 40K. Beyond this threshold, the performance of pre-trained models starts to fall behind its non-pretrained counterpart. This behavior suggests the phenomenon of *parameter ossification* [54] in pre-trained models, suggesting that pre-training can inadvertently "freeze" the model weights in a

---

[3]We confirm that the performance degradation is a common phenomenon rather than an outcome of limited model capacity by further altering the number of model layers. Readers can refer to Appendix C for more discussions.

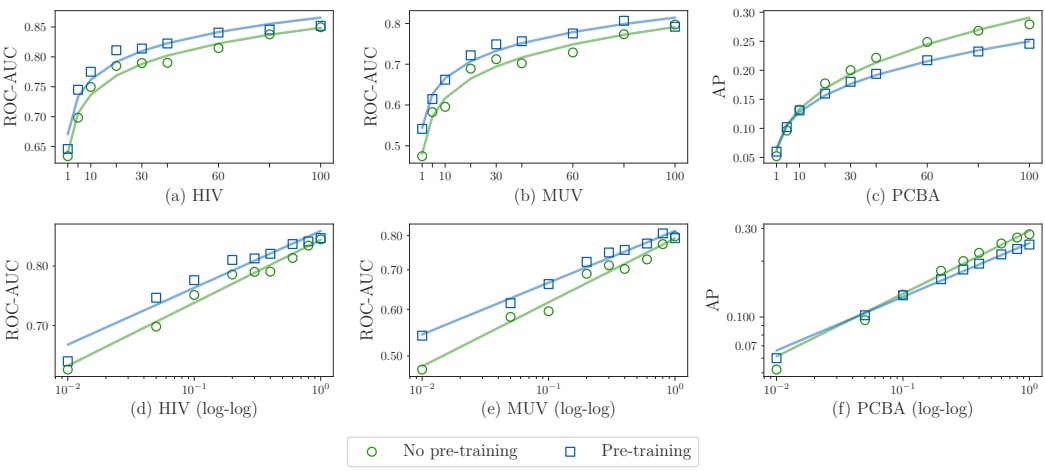

Figure 4: Comparison of scaling laws with and without pre-training on varied downstream data sizes.

way that limits their adaptability to the fine-tuning distribution, especially in large-data scenarios. Consequently, we believe the role and implications of pre-training in MRL warrant careful evaluation.

## 3.4 Influence of Dataset Splits on the Scaling Laws

The practice of dataset splitting in MRL is usually around two methodologies: random and scaffold splitting [38]. Random splitting ensures that both training and test samples are sourced from the same distribution, representing a uniform distribution setting. Conversely, scaffold splitting simulates the Out-Of-Distribution (OOD) scenario, wherein the training and test distributions are entirely distinct, pushing the boundaries of OOD generalization capabilities of MRL models. However, the realities of drug discovery often diverge from these two extremes. It is frequent for test compounds to possess substructures already encountered in the training samples [55]. To address this more realistic situation, we introduce imbalanced splitting, which generates the training, validation, and test sets based on scaffolds at first and then moves a fraction of the samples (5% in our experiments) from both test and validation sets to the training set. This approach serves as a trade-off between random and scaffold splitting, offering a more realistic reflection of real-world scenarios.

In this section, we still focus on 2D graph modalities and the learning behaviors on these three splitting methods by changing the scale of downstream datasets are shown in Figure 5. For readers interested in a deeper exploration, we append additional experiments in Appendix C.3 that analyze the neural scaling laws of two other modalities fingerprints and SMILES strings.

**Observation 4.** Our empirical observations validate that regardless of the dataset splitting methods, model performance conforms to a power law. Note that the performance curve with random splitting has the highest exponent, which suggests that this uniform setting has a higher data utilization efficiency compared to the other two OOD settings. We also observe outlier performance in the imbalanced splitting of the MUV dataset, where certain data scales (e.g., 80% and 100%) break the the power laws, which we suspect is due to the examples selected outside the training set could mislead the model to capture spurious relationship between label and features. Importantly, these findings underscore the inherent challenges of MRL in real-world scenarios, which often face with the pervasive issues of distribution shifts.

## 3.5 Influence of the Model Capacity on the Scaling Laws

This section we study the relationship between model performance and capacity. In our MRL models, the capacity is largely determined by the number of layers (depth $D$) and hidden units (width $W$). Therefore, we characterize model capacity as $D \times W$, since the parameters of the embedding layer and output layer are the same across different models and hence can be sidelined from our evaluation. In the experiments, we focus on 2D topology graphs with GIN as encoders. Due to the inherit drawbacks of message-passing networks, such as as over-smoothing and over-squashing, we carefully select four experimental configurations with reasonable increments: $[64 \times 2, 100 \times 3, 300 \times 5, 600 \times 10]$.

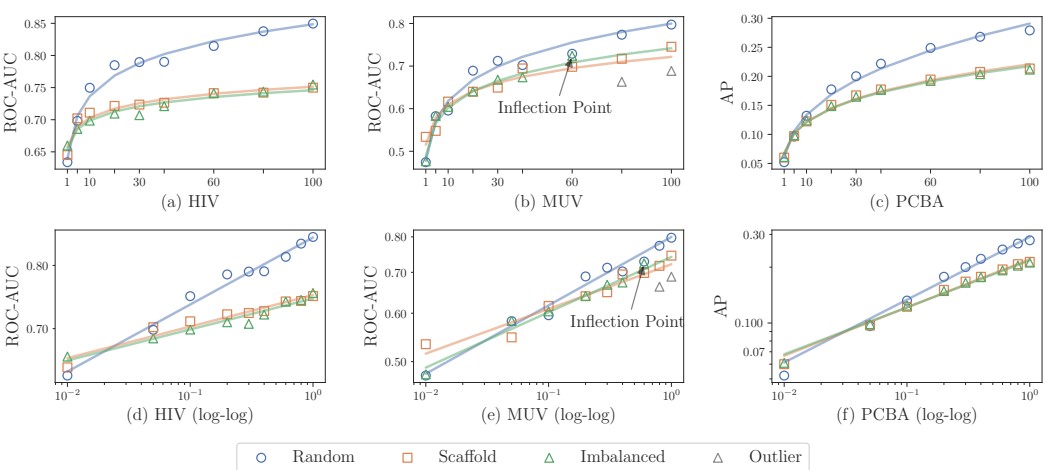

Figure 5: Neural scaling laws with three dataset splitting schemes: random, scaffold, and imbalanced splitting. Arrows indicate inflection points that deviate significantly from the fitted curves.

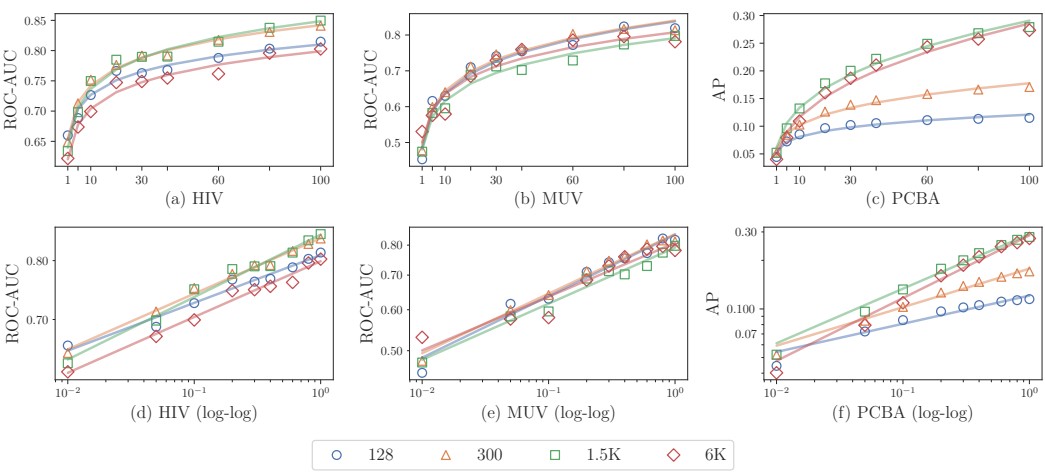

Figure 6: Neural scaling laws with the model capacity. We study four configurations of a GIN model with 128, 300, 1.5K, and 6K parameters respectively.

The performance with these four model configurations across different datasets is summarized in Figure 6.

**Observation 5.** The experimental results demonstrate that the relationship between model performance and capacity also adheres to the power law However, it is observed that the optimal model capacity varies across different tasks and that a smaller model could be sufficient to achieve the optimal performance for some tasks. For example, for HIV and PCBA datasets, the model with a capacity of 1.5K parameters achieves the best performance. On the contrary, the smallest model with a capacity of 128 parameters achieves optimal performance on the MUV dataset. An interesting observation is that, on the PCBA dataset with over 400K data samples, there is significant difference in power law exponents among different model capacity. The models with lower capacity achieve saturated performance quickly as the data scales up, whereas this phenomenon is not evident in HIV and MUV datasets. In summary, we believe it is important to carefully select the appropriate model capacity according to the characteristics of datasets and tasks.

## 3.6 Data Pruning for Molecular Representation Learning

Lastly, we investigate the data pruning strategies in MRL, which identify a representative subset of the complete dataset. Such strategies have been proved effective to improving training efficiency in

Table 1: Empirical performance of data pruning strategies on eight subsets of the PCBA dataset in terms of ROC-AUC (%, ↑). Results that are significantly higher or lower than random pruning (with a $p$-value of less than 5% in the significance test) are highlighted.

| *Uniform* | 1% | 5% | 10% | 20% | 30% | 40% | 60% | 80% |
|---|---|---|---|---|---|---|---|---|
| Random | $5.2_{\pm0.1}$ | $9.6_{\pm0.2}$ | $13.2_{\pm0.2}$ | $17.7_{\pm0.5}$ | $20.0_{\pm0.6}$ | $22.2_{\pm0.6}$ | $24.9_{\pm0.5}$ | $26.8_{\pm0.4}$ |
| Herding | $3.7_{\pm1.5}$ | $9.7_{\pm0.6}$ | $10.2_{\pm3.6}$ | $15.8_{\pm3.7}$ | $20.7_{\pm0.7}$ | $22.6_{\pm0.6}$ | $25.4_{\pm0.8}$ | $24.5_{\pm3.4}$ |
| Entropy | $5.4_{\pm0.2}$ | $10.0_{\pm0.4}$ | $13.7_{\pm0.6}$ | $17.6_{\pm0.1}$ | $20.2_{\pm0.6}$ | $22.1_{\pm0.4}$ | $25.0_{\pm0.3}$ | $26.6_{\pm0.4}$ |
| Least Confidence | $5.3_{\pm0.1}$ | $9.7_{\pm0.3}$ | $13.7_{\pm0.5}$ | $17.6_{\pm0.4}$ | $20.3_{\pm0.3}$ | $22.0_{\pm0.5}$ | $25.0_{\pm0.3}$ | $26.5_{\pm0.2}$ |
| Forgetting | $5.5_{\pm0.2}$ | $9.8_{\pm0.4}$ | $13.5_{\pm0.4}$ | $17.5_{\pm0.3}$ | $20.4_{\pm0.3}$ | $22.1_{\pm0.2}$ | $24.7_{\pm0.6}$ | $26.5_{\pm0.2}$ |
| GraNd | $5.5_{\pm0.3}$ | $9.7_{\pm0.3}$ | $13.3_{\pm0.4}$ | $17.5_{\pm0.4}$ | $20.3_{\pm0.5}$ | $22.1_{\pm0.5}$ | $24.9_{\pm0.3}$ | $26.8_{\pm0.3}$ |
| $k$-means | $4.1_{\pm0.2}$ | $8.1_{\pm0.3}$ | $11.6_{\pm0.3}$ | $16.5_{\pm0.2}$ | $20.4_{\pm0.4}$ | $22.6_{\pm0.3}$ | $24.9_{\pm0.4}$ | $26.3_{\pm0.1}$ |

| *Imbalanced* | 1% | 5% | 10% | 20% | 30% | 40% | 60% | 80% |
|---|---|---|---|---|---|---|---|---|
| Random | $6.1_{\pm0.2}$ | $9.9_{\pm0.2}$ | $12.4_{\pm0.5}$ | $14.9_{\pm0.3}$ | $16.5_{\pm0.2}$ | $17.7_{\pm0.2}$ | $19.2_{\pm0.1}$ | $20.4_{\pm0.2}$ |
| Herding | $4.7_{\pm0.6}$ | $8.5_{\pm0.5}$ | $11.5_{\pm0.3}$ | $13.4_{\pm1.6}$ | $14.7_{\pm2.8}$ | $16.5_{\pm2.1}$ | $16.8_{\pm3.6}$ | $17.9_{\pm3.1}$ |
| Entropy | $6.0_{\pm0.3}$ | $9.9_{\pm0.4}$ | $12.3_{\pm0.4}$ | $15.1_{\pm0.5}$ | $16.8_{\pm0.5}$ | $17.8_{\pm0.2}$ | $19.3_{\pm0.4}$ | $20.5_{\pm0.3}$ |
| Least Confidence | $6.0_{\pm0.3}$ | $9.9_{\pm0.2}$ | $12.3_{\pm0.4}$ | $15.3_{\pm0.3}$ | $17.0_{\pm0.5}$ | $17.9_{\pm0.3}$ | $19.4_{\pm0.3}$ | $20.5_{\pm0.1}$ |
| Forgetting | $5.7_{\pm0.3}$ | $9.9_{\pm0.2}$ | $12.2_{\pm0.3}$ | $14.7_{\pm0.2}$ | $15.8_{\pm0.3}$ | $17.8_{\pm0.2}$ | $19.3_{\pm0.5}$ | $20.3_{\pm0.3}$ |
| GraNd | $5.9_{\pm0.4}$ | $9.7_{\pm0.3}$ | $12.2_{\pm0.5}$ | $15.0_{\pm0.6}$ | $16.8_{\pm0.3}$ | $17.8_{\pm0.4}$ | $19.4_{\pm0.2}$ | $20.7_{\pm0.5}$ |
| $k$-means | $4.6_{\pm0.3}$ | $8.1_{\pm0.3}$ | $10.7_{\pm0.3}$ | $13.9_{\pm0.2}$ | $16.4_{\pm0.4}$ | $17.2_{\pm0.3}$ | $19.6_{\pm0.5}$ | $20.4_{\pm0.1}$ |

CV and NLP domains [35–37] but their roles in MRL remain under-explored. In our experiments, we benchmark seven data pruning strategies originally designed for image data and adapt them for 2D molecular graphs: Herding [56], Entropy [57], Least Confidence [57], Forgetting [58], GraNd [59], and $k$-means [60]. We also include random pruning as a baseline method. Please refer to Appendix C.7 for a detailed description of these adopted data pruning strategies.

For all experiments, we employ each data pruning strategy to curate eight subsets, which are constituted of the following percentages of the whole dataset: [1%, 5%, 10%, 20%, 30%, 40%, 60%, 80%]. For a comprehensive investigation, we compare the results with two data splitting schemes to discern if performance of pruning strategies might be influenced by data distribution disparities. Table 1 summarizes the results on PCBA, one of our largest datasets, where we highlight significant results with a $p$-value less than 5% in the significance test (t-test) compared with random pruning.

**Observation 6.** Our empirical analysis reveals that no adopted pruning strategy consistently outperforms random pruning across various data subsets, irrespective of the dataset splitting methodology. The performance difference among these pruning strategies also remain narrow. These trends are consistently observed across the MUV and HIV datasets as well, with details provided in Appendix C.7. This leads us to postulate two potential hypotheses. First, the current molecular encoders might be adept at extracting the information from the tested datasets, thereby suggesting a minimal presence of data redundancy. This suggests a potential need for significantly larger datasets to showcase the efficacy of data pruning methods in MRL. Second, existing data pruning strategies, originally proposed for image data, may not transfer their effectiveness to molecular data. For example, similarity-based approaches like Herding might need the infusion of domain-specific knowledge when measuring the similarity of molecular graphs.

## 4 Conclusions

In this paper, we conduct a series of data-centric experiments on Molecular Representation Learning (MRL) spanning four dimensions, including (1) data modality, (2) dataset splitting, (3) the role of pre-training and (4) model capacity, to investigate the model performance under different data scales. Our empirical results demonstrate that the scaling behaviors generally conform to a power law, implying the marginal effect of adding more molecular data. Moreover, we observe that there is no one-size-fits-all configuration to different MRL datasets, which calls for task-specific analysis in real-world application. We also benchmark seven data pruning strategies and observe that none of these methods outperform random sampling. Our analysis underscores the unique challenges inherent in MRL, such as distribution shift in molecular discovery and the non-Euclidean nature of molecular

datasets. We anticipate that our findings will catalyze further research, driving more data-efficient methodologies for molecular representation learning.

**Limitations.** We conduct our experiments using widely-adopted model architectures in the field, such as GIN for the 2D topology graph, SchNet for the 3D geometry graph, and Transformer for SMILES string and Fingerprint. However, the rapid development of molecular representation learning in recent years has led to the emergence of models with improved expressiveness. The neural scaling laws on these models is still to be explored. Moreover, our focus is primarily on investigating the impact of various dimensions on supervised molecular representation learning. Despite that we have considered the effect of pre-training, we do not explore the neural scaling behaviors between data scales in pre-training and the corresponding performance on the downstream tasks.

**Future work.** In closing, we point out several prospective avenues for further exploration. (1) Neural scaling laws in molecular generative tasks. In this paper, our exploration has concentrated solely on predictive tasks, leaving an unexplored domain of generative tasks. Potential research might investigate neural scaling laws on tasks such as generation of drugs, conformations, and docking poses, to name a few. (2) Refinement of pre-training strategies. In our experiments, we discover that pre-training on molecular data does not improve efficiency of leveraging downstream data. Future work could consider to address the issues of parameter ossification to improve data efficiency in pre-trained MRL models. (3) In-depth analysis of data pruning strategies for molecular data. We observe that the existing image pruning methods do not significantly improve MRL performance. Such observations call for a rigorous evaluation of data pruning methodologies, potentially on more larger-scale datasets, to ascertain the potential advantages of data pruning techniques. Moreover, the design of efficient pruning strategies specifically tailored to molecular data is also a promising direction, which yet remain largely unexplored.

# Acknowledgements

The authors would like to thank anonymous reviewers for their helpful feedback. This work is supported by National Natural Science Foundation of China (62141608, 62206291, 62372454).

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

# A    Implementation Details

## A.1    Molecular Encoders

In this section, we detail the implementation of each encoder. We denote the representation for node (atom) $v_i$ as $\boldsymbol{h}_i$ and the representation at the graph (molecule) level as $\boldsymbol{z}$.

**Embedding 2D graphs.**    Graph Isomorphism Network (GIN) [5] is a simple and effective model to learn discriminative graph representations, which is proved to have the same representational power as the Weisfeiler-Lehman test [61]. Recall that each molecule is represented as $\mathcal{G} = (\boldsymbol{A}, \boldsymbol{X}, \mathsf{E})$, where $\boldsymbol{A}$ is the adjacency matrix, $\boldsymbol{X}$ and $\mathsf{E}$ are features for atoms and bonds respectively. The layer-wise propagation rule of GIN can be written as:

$$\boldsymbol{h}_i^{(k+1)} = f_{\text{atom}}^{(k+1)} \left( \boldsymbol{h}_i^{(k)} + \sum_{j \in \mathcal{N}(i)} \left( \boldsymbol{h}_j^{(k)} + f_{\text{bond}}^{(k+1)}(\boldsymbol{E}_{ij}) \right) \right), \tag{2}$$

where the input features $\boldsymbol{h}_i^{(0)} = \boldsymbol{x}_i$, $\mathcal{N}(i)$ is the neighborhood set of atom $v_i$, and $f_{\text{atom}}$, $f_{\text{bond}}$ are two MultiLayer Perceptron (MLP) layers for transforming atoms and bonds features, respectively. By stacking $K$ layers, we can incorporate $K$-hop neighborhood information into each center atom in the molecular graph. Then, we take the output of the last layer as the atom representations and further use the mean pooling to get the graph-level molecular representation:

$$\boldsymbol{z}^{\text{2D}} = \frac{1}{N} \sum_{i \in \mathcal{V}} \boldsymbol{h}_i^{(K)}. \tag{3}$$

**Embedding 3D graphs.**    We use the SchNet [43] as the encoder for the 3D geometry graphs. SchNet models message passing in the 3D space as continuous-filter convolutions, which is composed of a series of hidden layers, given as follows:

$$\boldsymbol{h}_i^{(k+1)} = f_{\text{MLP}} \left( \sum_{j=1}^{N} f_{\text{FG}}(\boldsymbol{h}_j^{(t)}, \boldsymbol{r}_i, \boldsymbol{r}_j) \right) + \boldsymbol{h}_i^{(t)}, \tag{4}$$

where the input $\boldsymbol{h}_i^{(0)} = \boldsymbol{a}_i$ is an embedding dependent on the type of atom $v_i$, $f_{\text{FG}}(\cdot)$ denotes the filter-generating network. To ensure rotational invariance of a predicted property, the message passing function is restricted to depend only on rotationally invariant inputs such as distances, which satisfying the energy properties of rotational equivariance by construction. Moreover, SchNet adopts radial basis functions to avoid highly correlated filters. The filter-generating network is defined as follow:

$$f_{\text{FG}}(\boldsymbol{x}_j, \boldsymbol{r}_i, \boldsymbol{r}_j) = \boldsymbol{x}_j \cdot e_k(\boldsymbol{r}_i - \boldsymbol{r}_j) = \boldsymbol{x}_j \cdot \exp(-\gamma \|\|\boldsymbol{r}_i - \boldsymbol{r}_j\|_2 - \mu\|_2^2). \tag{5}$$

Similarly, for non-quantum properties prediction concerned in this work, we take the average of the node representations as the 3D molecular embedding:

$$\boldsymbol{z}^{\text{3D}} = \frac{1}{N} \sum_{i \in \mathcal{V}} \boldsymbol{h}_i^{(K)}, \tag{6}$$

where $K$ is the number of hidden layers.

**Embedding fingerprints & SMILES strings.**    Due to the discrete and extremely sparse nature of fingerprint vectors, we first transform all $F$ binary feature fields into a dense embedding matrix $\boldsymbol{F}^{fp} \in \mathbb{R}^{F^{fp} \times D_{\text{F}}}$ via embedding lookup, while we transform SMILES tokens in the same way via another embedding lookup $\boldsymbol{F}^{sm} \in \mathbb{R}^{F^{sm} \times D_{\text{F}}}$. Then, we introduce a positional embedding matrix $\boldsymbol{P} \in \mathbb{R}^{F \times D_{\text{F}}}$ to capture the positional relationship among bits in the fingerprint vector, which is defined as:

$$\boldsymbol{P}_{p,2i} = \sin(p/10000^{2i/D_F}), \tag{7}$$

$$\boldsymbol{P}_{p,2i+1} = \cos(p/10000^{2i/D_F}), \tag{8}$$

where $p$ denotes the corresponding bit position and $i$ is corresponds to the $i$-th embedding dimension. The positional embedding matrix will be added to the transformed embedding matrix:

$$\boldsymbol{F} = \boldsymbol{F}' + \boldsymbol{P}. \tag{9}$$

Thereafter, we use a multihead Transformer [45] to model the interaction among those feature fields. Specifically, we first transform each feature into a new embedding space as:

$$\boldsymbol{Q}^{(h)} = \boldsymbol{F}\boldsymbol{W}_{\mathrm{Q}}^{(h)}, \tag{10}$$

$$\boldsymbol{K}^{(h)} = \boldsymbol{F}\boldsymbol{W}_{\mathrm{K}}^{(h)}, \tag{11}$$

$$\boldsymbol{V}^{(h)} = \boldsymbol{F}\boldsymbol{W}_{\mathrm{V}}^{(h)}, \tag{12}$$

where the three linear transformation matrices $\boldsymbol{W}_{\mathrm{Q}}^{(h)}, \boldsymbol{W}_{\mathrm{K}}^{(h)}, \boldsymbol{W}_{\mathrm{V}}^{(h)} \in \mathbb{R}^{D_{\mathrm{F}} \times D/H}$ parameterize the query, key, and value transformations for the $h$-th attention head, respectively. Following that, we compute the attention scores among all feature pairs and then linearly combine the value matrix from all $H$ attention heads:

$$\boldsymbol{W}_{\mathrm{A}}^{(h)} = \mathrm{softmax}\left(\frac{\boldsymbol{Q}^{(h)}(\boldsymbol{K}^{(h)})^{\top}}{\sqrt{D_{\mathrm{H}}}}\right), \tag{13}$$

$$\widehat{\boldsymbol{Z}} = \left[\boldsymbol{W}_{\mathrm{A}}^{(1)}\boldsymbol{V}^{(1)}; \boldsymbol{W}_{\mathrm{A}}^{(2)}\boldsymbol{V}^{(2)}; \ldots; \boldsymbol{W}_{\mathrm{A}}^{(H)}\boldsymbol{V}^{(H)}\right], \tag{14}$$

Finally, we perform sum pooling on the resulting embedding matrix $\widehat{\boldsymbol{Z}} \in \mathbb{R}^{F \times D_{\mathrm{F}}}$ and use a linear model $f_{\mathrm{LIN}}$ to obtain the final fingerprint or SMILES string embedding $\boldsymbol{z} \in \mathbb{R}^{D}$:

$$\boldsymbol{z} = f_{\mathrm{LIN}}\left(\sum_{d=1}^{D_{\mathrm{F}}} \widehat{\boldsymbol{Z}}_d\right). \tag{15}$$

### A.2 Computing infrastructures

**Software infrastructures.** All of the experiments are implemented in Python 3.7, with the following supporting libraries: PyTorch 1.10.2 [62], PyG 2.0.3 [63], RDKit 2022.03.1 [64] and HuggingFace's Transformers 4.17.0 [65].

**Hardware infrastructures.** We conduct all experiments on a computer server with 8 NVIDIA GeForce RTX 3090 GPUs (with 24GB memory each) and 256 AMD EPYC 7742 CPUs.

### A.3 Code availability

The source code of our empirical implementation can be accessed at `https://github.com/Data-reindeer/NSL_MRL`.

## B  Datasets and Tasks

In the following, we will elaborate on the adopted datasets and the statistics are summarized in Table 2.

**Datasets.** We consider four datasets ranging from molecular-level properties to macroscopic influences on human body for experimental investigation: HIV [39], MUV [40], PCBA [41] and QM9 [42].

- HIV (AIDS Antiviral Screen) was developed by the Drug Therapeutics Program (DTP) [39], which is designed to evaluate the ability of molecular compounds to inhibit HIV replication.

- Maximum Unbiased Validation (MUV) group was selected from PubChem BioAssay via a refined nearest neighbor analysis approach, which is specifically designed for validation of virtual screening techniques [40].

- PubChem BioAssay (PCBA) is a database consisting of biological activities of small molecules generated by high-throughput screening [41].

Table 2: Statistics of datasets used in experiments.

| | Dataset | Data Type | #Molecules | Avg. #atoms | Avg. #bonds | #Tasks | Avg. degree |
|---|---|---|---|---|---|---|---|
| Pre-training | PCQM4Mv2 | SMILES | 3,746,620 | 14.14 | 14.56 | - | 2.06 |
| Classification | MUV | SMILES | 93,087 | 24.23 | 26.28 | 17 | 2.17 |
| | HIV | SMILES | 41,127 | 25.51 | 27.47 | 1 | 2.15 |
| | PCBA | SMILES | 437,929 | 25.96 | 28.09 | 92 | 2.16 |
| Regression | QM9-$\epsilon_{gap}$ | SMILES, 3D | 130,831 | 18.03 | 18.65 | 1 | 2.07 |
| | QM9-U0 | SMILES, 3D | 130,831 | 18.03 | 18.65 | 1 | 2.07 |
| | QM9-ZPVE | SMILES, 3D | 130,831 | 18.03 | 18.65 | 1 | 2.07 |

- QM9 is a comprehensive dataset that provides geometric, energetic, electronic and thermodynamic properties for a subset of GDB-17 database, comprising 134 thousand stable organic molecules with up to nine heavy atoms [42]. In our experiments, we delete 3,054 uncharacterized molecules which failed the geometry consistency check [66]. We include the $\epsilon_{gap}$, U0, and ZPVE in our experiment, which cover properties related to electronic structure, stability, and thermodynamics. These properties collectively capture important aspects of molecular behavior and can effectively represent various energetic and structural characteristics within the QM9 dataset.

- PCQM4Mv2 is a quantum chemistry dataset originally curated under the PubChemQC project [49]. Based on the PubChemQC, Hu et al. [50] define a meaningful ML task of predicting DFT-calculated HOMO-LUMO energy gap of molecules given their 2D molecular graphs. The HOMO-LUMO gap is one of the most practically-relevant quantum chemical properties of molecules since it is related to reactivity, photoexcitation, and charge transport.

## C    Additional Experimental Results

### C.1    The Neural Scaling Law on PCQM4Mv2 Dataset

Following the experimental settings of Section 3.1, we conduct additional experiments on the PCQM4Mv2, a large-scale challenge dataset provided by OGB [4]. To the best of our knowledge, it is currently the largest labeled molecular dataset available, containing over 3.7 million molecular samples. It is worth noting that due to the unavailability of official test labels, we selected 50% from the validation set as the test data and report the results in Figure 7.

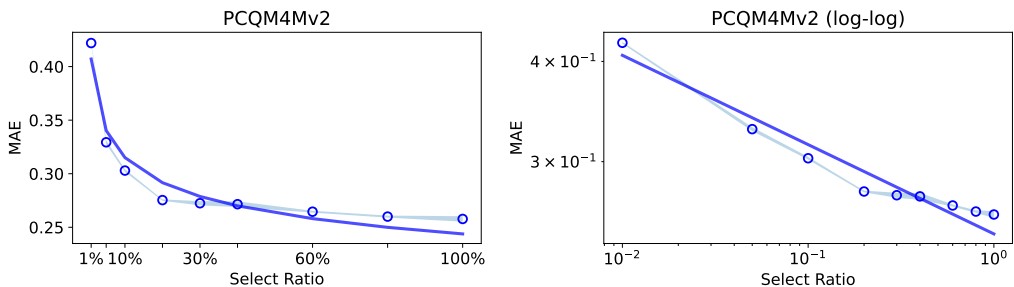

Figure 7: The neural scaling law of molecular representation learning on PCQM4Mv2 dataset.

### C.2    The Effect of Equivariant Representations on the Scaling Law

To further explore the scaling laws of more powerful 3D graph representations, we have extended our investigation to include SE(3) invariant model (SphereNet) and E(3) equivariant model (PaiNN) in the context of Section 3.1. The empirical results are demonstrated in Figure 8. It can be observed that most of the performances generally adhere to the power-law relationship and the difference lies in the coefficients and exponents of the power law. Furthermore, the relative differences in performance

---

[4]https://ogb.stanford.edu/docs/lsc/pcqm4mv2

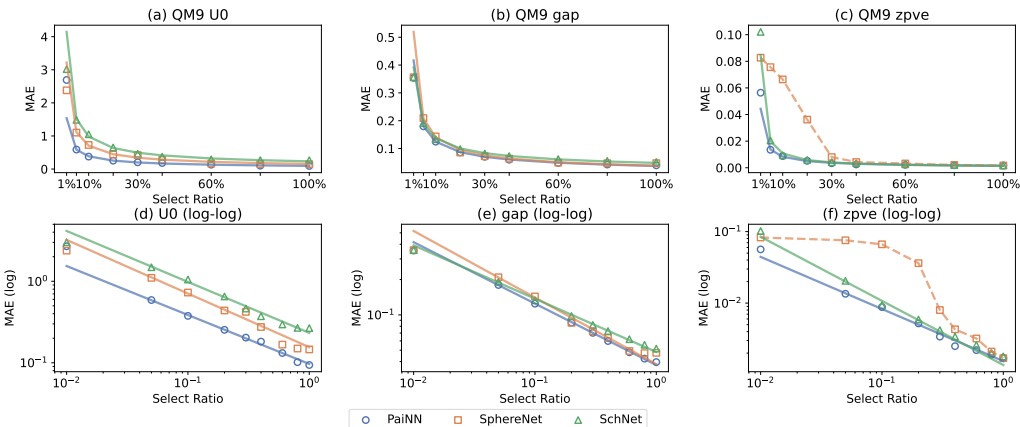

Figure 8: The neural scaling law of equivariant representations with 3D graph modality.

between the three models tend to vary with distinct predictive tasks. For instance, in the U0 prediction task, the power law relationships of the three models exhibit a noticeable parallel pattern in the log-log plot, whereas in the other two tasks, a crossing effect becomes evident.

## C.3 More Results of the Effect of Data Split

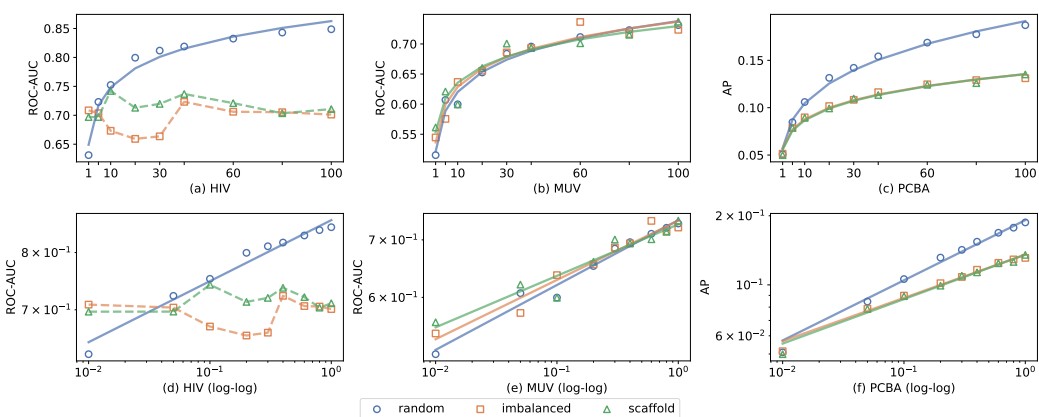

Figure 9: The effect of data split on the scaling law with fingerprint modality.

To enhance the robustness of our findings in Section 3.4, in addition to the comparison of the 2D graph modality across various data splits provided in the main text, we also conduct analogous experiments on the fingerprint and SMILES modalities. The empirical results, illustrated in Figure 9 and Figure 10, align with our observations in the Section 3.4: (1) Most performance within these modalities adhere to the power law relationship. (2) The SMILES modality still exhibits a performance drop phenomenon in the MUV dataset. The only deviation lies in the smallest dataset, HIV, where the performance variation under imbalanced/scaffold split settings does not conform to the power law. This deviation might be attributed to the small scale of the HIV dataset, posing challenges for models to generalize and extrapolate effectively.

## C.4 Comparison of Different Fingerprint Encoders

Given the absence of established encoder design for fingerprints in the field, we conduct a comparison between two classic encoders, a single-layer MLP and a single-layer Transformer, in terms of their performance variation. The results indicate that both encoders exhibit a strong adherence to the

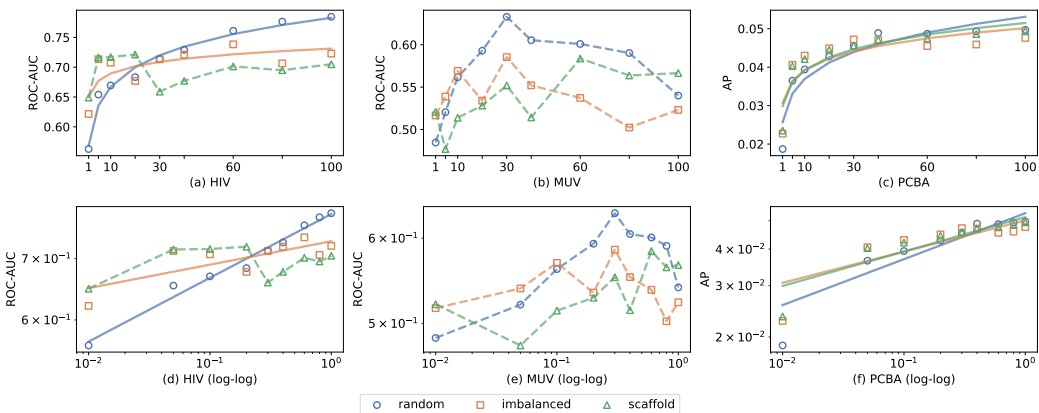

Figure 10: The effect of data split on the scaling law with SMILES modality.

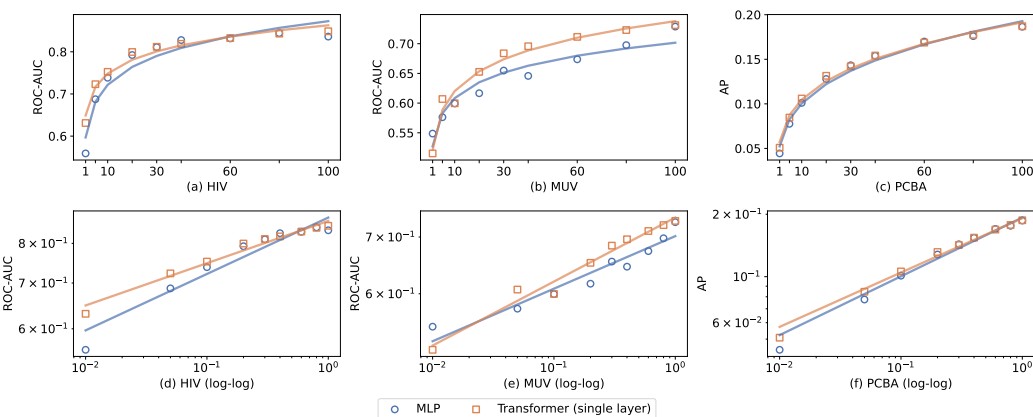

Figure 11: The Effect of Model Architecture on the Scaling Law with fingerprint Modality.

power law relationship in their scaling behavior. As such, the choice of encoder does not impact the robustness of our conclusions.

## C.5 Single-property performance of MUV dataset with SMILES modality

In order to gain a more detailed understanding of the performance degradation phenomenon in the multitask scenario of MUV, we specifically demonstrate the neural scaling behavior of the SMILES modalities in single-property ROC-AUC, as shown in Figure 12. Despite some ascending behavior, the bulk of properties exhibit varying degrees of performance drop in the last few proportions, which accounts for the overall performance drop in the multi-task setting.

## C.6 Results of different-layers Transformer with SMILES modality.

Figure 13 presents the scaling behavior of transformers with different numbers of layers in terms of their performance. It can be observed that RoBERTa achieves the overall best performance, followed by a 1-layer Transformer, and the worst performance is exhibited by a 3-layer one. In contrast to fingerprint, the SMILES modality could experience a performance drop on some datasets (HIV and MUV) in the high-data regime. Additionally, the varying numbers of Transformer layers do not affect our conclusions in Section 3.2 regarding modality comparison, as even the superior RoBERTa model overall does not surpass the performance of graph and fingerprint modalities.

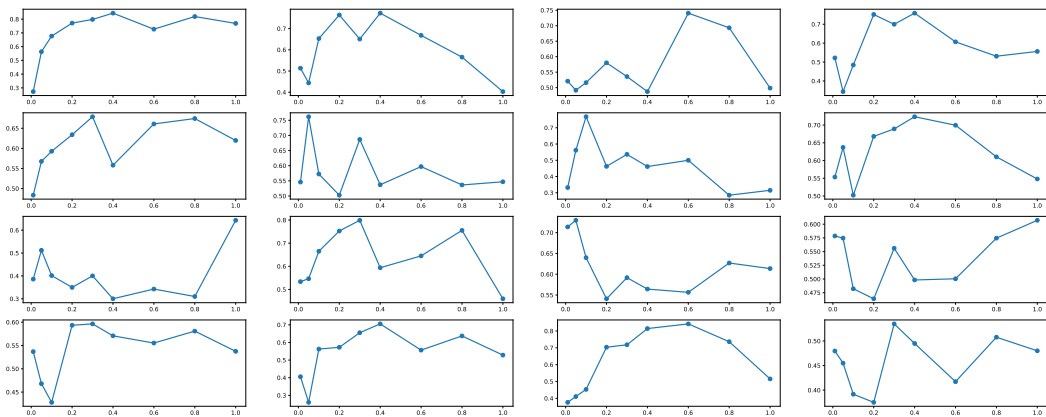

Figure 12: The neural scaling law of single-property performance (ROC-AUC) of MUV dataset with SMILES modality.

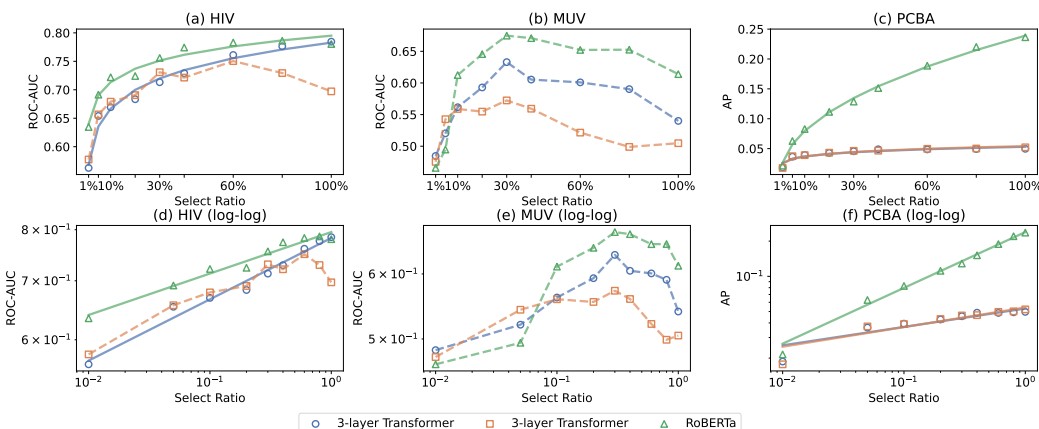

Figure 13: The neural scaling law of different-layer model (Transformer) performance with SMILES modality.

## C.7 Data pruning strategies and additional results

**Problem statement.** Consider a learning scenario where we have a large training set denoted as $\mathcal{T} = (\boldsymbol{x_i}, y_i)_{i=1}^{|\mathcal{T}|}$, consisting of input-output pairs $(\boldsymbol{x_i}, y_i)$, where $\boldsymbol{x_i} \in \mathcal{X}$ represents the input and $y_i \in \mathcal{Y}$ denotes the ground-truth label corresponding to $x_i$. Here, $\mathcal{X}$ and $\mathcal{Y}$ refer to the input and output spaces, respectively. The objective of data pruning is to identify a subset $\mathcal{S} \subset \mathcal{T}$, satisfying the constraint $|\mathcal{S}| < |\mathcal{T}|$, that captures the most informative instances. This subset, when used to train a model denoted as $\boldsymbol{\theta}^{\mathcal{S}}$, should yield a similar or better generalization performance to that of the model $\boldsymbol{\theta}^{\mathcal{T}}$, which is trained on the entire training set $\mathcal{T}$.

**Data pruning strategies.** In our data pruning experiments, we implement a total of seven data pruning (or coreset selection) strategies: Herding [56], Entropy [57], Least Confidence [57], Forgetting [58], GraNd [59], K-means [60] and we additionally include random pruning as a baseline method. These seven strategies are widely used in the field of CV [67] and have potential in mitigating the issue of data redundancy in large-scale datasets, thereby saving computational and storage resources. Here we provide a brief overview to each of them.

- **Herding** [56] operates by selecting data points in the feature space based on the distance between the coreset center and the original center. It follows an incremental and greedy approach, adding one sample at a time to the coreset in order to minimize the distance between the centers.

Table 3: Empirical performance of data pruning strategies on eight subsets of the MUV dataset in terms of ROC-AUC (%, ↑). Results that are significantly higher or lower than random pruning (with a $p$-value of less than 5% in the significance test) are highlighted.

| *Uniform* | 1% | 5% | 10% | 20% | 30% | 40% | 60% | 80% |
|---|---|---|---|---|---|---|---|---|
| Random | 47.4±3.5 | 58.3±3.4 | 59.6±4.7 | 68.9±2.1 | 71.2±3.4 | 70.2±3.0 | 72.9±1.7 | 77.4±2.8 |
| Herding | 48.4±4.7 | 51.9±4.8 | 53.1±10.3 | 61.5±4.9 | 68.5±7.3 | 65.1±5.6 | 71.7±6.4 | 78.7±5.0 |
| Entropy | 48.9±5.5 | 58.5±5.1 | 62.8±4.0 | 63.8±4.2 | 68.7±3.4 | 71.0±3.7 | 75.0±2.5 | 79.0±3.8 |
| Least Confidence | 52.0±7.9 | 57.4±5.7 | 61.1±3.3 | 66.3±2.1 | 67.6±4.8 | 70.0±3.6 | 75.1±2.5 | 78.5±2.2 |
| Forgetting | 47.1±2.1 | 58.6±2.1 | 60.9±5.2 | 63.3±1.8 | 67.1±3.0 | 70.5±2.8 | 74.5±2.6 | 76.9±3.8 |
| GraNd | 47.3±3.4 | 52.2±7.1 | 64.7±5.1 | 65.9±3.1 | 64.4±5.9 | 71.4±4.4 | 72.5±2.5 | 76.2±3.2 |
| *k*-means | 49.9±4.7 | 60.5±7.8 | 65.5±3.7 | 68.0±4.2 | 65.5±3.7 | 67.1±2.8 | 72.1±1.4 | 76.9±4.0 |

| *Imbalanced* | 1% | 5% | 10% | 20% | 30% | 40% | 60% | 80% |
|---|---|---|---|---|---|---|---|---|
| Random | 47.7±3.5 | 58.3±3.4 | 60.4±4.7 | 64.1±2.1 | 66.8±3.4 | 67.4±3.0 | 72.4±1.7 | 66.3±2.8 |
| Herding | 50.2±4.7 | 55.8±4.8 | 59.3±10.3 | 58.8±4.9 | 64.5±7.3 | 65.6±5.6 | 65.8±6.4 | 69.6±5.0 |
| Entropy | 47.7±5.5 | 57.5±5.1 | 61.0±4.0 | 68.0±4.2 | 64.4±3.4 | 67.2±3.7 | 68.1±2.5 | 69.5±3.8 |
| Least Confidence | 50.4±7.9 | 52.9±5.7 | 60.5±3.3 | 64.2±2.1 | 65.5±4.8 | 68.3±3.6 | 69.4±2.5 | 66.4±2.2 |
| Forgetting | 49.3±2.1 | 51.4±2.1 | 58.7±5.2 | 64.0±1.8 | 65.7±3.0 | 66.5±2.8 | 67.7±2.6 | 68.8±3.8 |
| GraNd | 52.0±3.4 | 55.3±7.1 | 63.8±5.1 | 63.4±3.1 | 67.0±5.9 | 68.6±4.4 | 70.3±2.5 | 69.0±3.2 |
| *k*-means | 52.7±4.7 | 56.0±7.8 | 58.5±3.7 | 61.0±4.2 | 63.4±3.7 | 63.4±2.8 | 67.1±1.4 | 70.0±4.0 |

- **Entropy** and **Least Confidence** [57] iteratively select samples with lower entropy and least confidence, respectively. These methods identify informative samples by considering that lower uncertainty can provide more information gain, thereby benefiting model training and reducing data redundancy.

- **Forgetting** [58] calculates the frequency of forgetting that occurs during the training process, which refers to the number of times the samples correctly classified in the previous epoch are misclassified in the current epoch. Those unforgettable samples, exhibiting robust performance across epochs, have minimal impact on model performance when removed.

- **GraNd** [59] measures the average impact of each sample on the reduction of training loss during the initial epochs. Training samples are more important if they contribute more to the error or loss when training neural networks.

- **$k$-means** [60] employs the application of $k$-means clustering in the latent space to define the difficulty of each data point based on its Euclidean distance to its nearest cluster centroid. Simple samples (with low difficulty) are considered for removal to reduce data redundancy. It is noteworthy that this method, unlike the aforementioned approaches, does not require any label information or training and can be directly applied to the dataset.

**Additional data pruning results.** Here we include additional results of data pruning performance on MUV and HIV datasets, as shown in Table 3 and Table 4. Observing the performance of these data pruning strategies, none of these methods consistently outperform random sampling, which aligns with our observation in Section 3.6. Note that in the case of MUV with an imbalanced data split, several data pruning methods such as Random, GraNd, and Least Confidence even demonstrate a performance decline when the subset scale surpasses 60%.

# D    Related Work

The following section provides a more broad literature review across the spectrum of molecular representation learning and neural scaling law.

## D.1    Molecular representation learning

The past decade has seen remarkable success in the application of deep learning in a variety of biochemical tasks, spanning from virtual screening [68] to molecular property prediction [6]. Within

Table 4: Empirical performance of data pruning strategies on eight subsets of the HIV dataset in terms of ROC-AUC (%, ↑). Results that are significantly higher or lower than random pruning (with a $p$-value of less than 5% in the significance test) are highlighted.

| Uniform | 1% | 5% | 10% | 20% | 30% | 40% | 60% | 80% |
|---|---|---|---|---|---|---|---|---|
| Random | 63.4±2.8 | 69.8±2.2 | 75.0±2.7 | 78.5±1.2 | 79.0±2.2 | 79.0±1.3 | 81.5±1.7 | 83.8±0.8 |
| Herding | 60.2±3.9 | 63.3±3.8 | 64.7±5.0 | 69.5±5.4 | 71.8±7.0 | 75.8±6.6 | 80.0±2.8 | 82.6±1.2 |
| Entropy | 67.9±2.2 | 71.1±3.7 | 74.2±1.6 | 76.2±1.2 | 77.0±2.0 | 79.2±1.8 | 81.4±1.9 | 83.2±1.4 |
| Least Confidence | 66.2±4.0 | 70.4±2.1 | 72.8±3.9 | 76.7±2.3 | 78.0±1.0 | 81.0±1.4 | 81.6±1.6 | 83.3±0.6 |
| Forgetting | 67.7±1.2 | 75.2±1.3 | 75.1±1.9 | 76.2±1.7 | 80.0±1.8 | 79.8±1.6 | 82.8±1.0 | 83.7±1.4 |
| GraNd | 66.2±4.0 | 69.3±2.6 | 73.6±2.0 | 78.1±1.1 | 78.1±1.6 | 78.6±1.0 | 82.3±0.8 | 83.2±1.4 |
| $k$-means | 63.8±4.8 | 64.4±3.4 | 65.7±1.8 | 68.1±1.6 | 71.5±1.4 | 72.5±3.5 | 79.2±0.5 | 82.3±2.2 |

| Imbalanced | 1% | 5% | 10% | 20% | 30% | 40% | 60% | 80% |
|---|---|---|---|---|---|---|---|---|
| Random | 66.6±1.7 | 68.6±3.1 | 69.9±3.7 | 70.9±1.4 | 70.7±4.1 | 72.1±3.0 | 74.1±1.1 | 74.4±1.1 |
| Herding | 57.1±3.0 | 63.0±3.8 | 64.9±3.6 | 65.8±5.9 | 67.3±6.0 | 72.6±1.8 | 73.3±2.2 | 73.7±0.6 |
| Entropy | 67.7±7.5 | 71.5±2.8 | 70.1±1.1 | 71.2±2.1 | 73.2±2.3 | 71.7±2.6 | 74.7±1.3 | 74.8±1.0 |
| Least Confidence | 66.8±5.2 | 71.4±1.0 | 71.3±2.6 | 71.8±2.7 | 69.5±2.8 | 73.7±3.4 | 73.4±2.6 | 73.8±1.8 |
| Forgetting | 66.1±3.1 | 69.7±5.8 | 70.2±3.6 | 71.9±1.9 | 71.6±1.8 | 71.4±2.0 | 73.9±1.4 | 74.2±2.3 |
| GraNd | 62.7±4.5 | 71.0±2.6 | 69.2±3.6 | 73.1±1.9 | 70.0±3.4 | 72.9±3.0 | 74.4±1.8 | 75.9±1.2 |
| $k$-means | 67.9±1.8 | 65.4±3.2 | 65.0±1.9 | 67.1±4.3 | 69.1±4.0 | 68.5±4.3 | 72.8±1.2 | 74.4±1.7 |

this context, molecular representation learning (MRL) serves as a pivotal link between the molecular modalities and the target tasks, efficiently capturing and encoding rich chemical semantic information into vector representations.

One of the mainstream research approaches in MRL is based on *2D topology graphs*. The advancements in Graph Neural Networks (GNNs) have enabled the application of more powerful GNN models in the field of molecular chemistry [5, 8, 11, 9, 10, 14], which has proven effective in enhancing the discriminability between representations and capturing underlying chemical semantics. The study of the expressive power of GNNs using the Weisfeiler-Lehman graph isomorphism test has been widely applied in MRL. GIN [5], as one of the most representative works, develops a simple and effective architecture based on a multi-perceptron layer (MLP) that has been proven to be as powerful as the WL test. Some works propose improvements in the expressive power of GNNs to address issues related to long-range interactions [11, 10], higher-order structures [9, 8] and substructure recognition [14] from different perspectives. Unlike traditional message passing mechanisms, Graphormer [12] have explored the direct application of Transformers [45] to graph representation with tailor-made positional encoding. A few research [13, 32] focus more on the ad-hoc model design for biochemical tasks, incorporating constraints based on molecular physics and chemical properties.

More recently, 3D graph representation for molecules has been largely explored thanks to its importance in modeling the dynamical behaviors of molecular structures [42, 69]. Different from 2D graph representation, euclidean group symmetry (E(3)/SE(3)) including translation, rotation and reflection needs to be baked into the design of the models. Specifically, SchNet [43] is one of the earliest work that incorporates euclidean distances between each pair of nodes as features which make the model E(3)-invariant. In addition to pairwise distance features, angle (between three atoms) and dihedral angle (between four atoms) are later on introduced as features into E(3)/SE(3)-invariant models [52, 70, 71]. Instead of invariant models, another branch of work explore equivariant design of 3D graph neural networks. EGNN [72] devises a simple equivariant update layer such that only linear transformation is applied to vector input to achieve equivariance in which similar ideas have also been applied to GVP [73] and PaiNN [51]. ClofNet [74] and LEFTNet [75] on the other side improves the expressiveness of such design by building local frams to scalarize equivariant inputs from there any neural architecture could be applied then transform back to vectors through a vectorization block without information loss. Another noticeable line of work leverage spherical harmonics and tensor product to construct equivariant layers [76–80].

In the advancements of supervised MRL, there has been limited progress in the model designs specifically tailored to the *SMILES string* and *fingerprint* modalities. It is worth noting that the

early benchmark models proposed in MoleculeNet [38] have maintained their competitiveness over time. With the rise of pre-training research paradigms, there has been promising progress in recent years towards pre-training approaches based on these two modalities [27, 28, 26, 16] as well as the former two. By employing contrastive [29–32] and generative [33, 30, 34] self-supervised strategies, molecular pre-training approaches guide the model training and subsequently facilitate positive transfer to downstream tasks. However, as mentioned in Section 3.3, existing molecular pre-training still suffer from issues such as parameter ossification [54], necessitating further exploration for more data-efficient and training-efficient models.

### D.2   Neural scaling law

The study of neural scaling law can be traced back to early theoretical analyses of bounding generalization error [81–84]. These works, based on assumptions about model capacity and data volume, reveal power-law relationships between the bounds of model generalization error and the amount of data. However, the conclusions drawn from these theoretical studies often yield loose or even vacuous bounds, leading to a disconnection between the theoretical findings and the empirical results of generalization error.

Early follow-on research have investigated empirical generalization error scaling, which represents an initial attempt at exploring the neural scaling law. Bango and Bill [85] conduct experiments on a language modeling problem called confusion set disambiguation, using subsets of a large-scale text corpus containing billions of words. Their findings suggest a power-law relationship between the average disambiguation validation error and the size of the training data. Similarly, Sun et al. [21] demonstrate that the accuracy of image classification models improves with larger data sizes and conclude that the accuracy increases logarithmically based on the volume of the training data size.

Hestness et al. [18] empirically validate that model accuracy improves as a power-law as growing training sets in various domains, which exhibit consistent learning behavior across model architectures, optimizers and loss functions. However, there exists generalization error plateau in small data region and irreducible error region. With a broader coverage, Michael et al. [22] present findings that consistently show the scaling behavior of language model log-likelihood loss in relation to non-embedding parameter count, dataset size, and optimized training computation. They leverage these relationships to derive insights into compute scaling, the extent of overfitting, early stopping step, and data requirements in the training of large language models.

In recent years, several investigations of neural scaling laws specific to particular tasks have been conducted [86, 19, 87, 60]. Unlike previous research, while power-law relationships hold within specific ranges of data size or model parameter count, certain tasks exhibit unique and uncommon learning behaviors. For instance, only marginal performance gains are expected beyond a few thousand examples in image reconstruction [20]. Moreover, there is a relevant study sharing a common spirit with our work in analyzing MRL from a scaling behavior viewpoint [88]. However, our research diverges in the following aspects: (1) *Scope of Neural Scaling Law Investigation*: The study of Frey et al. primarily considers large language models for generative chemical modeling and quantum property tasks. In contrast, our work encompasses a broader range of dimensions, such as modalities, data distributions, pre-training and model capacity, to investigate the influence of various factors on neural scaling laws. (2) *Different contributions*: Their contribution primarily revolves around developing strategies for scaling deep chemical models, notably large language models, and introducing novel model architectures. In contrast, our focus lies in exploring the impact of different dimensions on data utilization efficiency through a data-centric perspective and benchmarking existing data pruning strategies on MRL tasks.

