# OpenReview forum: "Uncovering Neural Scaling Laws in Molecular Representation Learning"
_NeurIPS.cc/2023/Track/Datasets_and_Benchmarks — NeurIPS 2023 Datasets and Benchmarks Poster_

### Official Review · Reviewer_jova · 2023-07-05
**Helpful for the AI4Science community**

**Rating:** 7
**Confidence:** 3

**Strengths:**

1. While research on molecular representation learning (MRL) is progressing at an amazing pace, the neural scaling laws specific to MRL have not received the same level of attention as other domains such as computer vision (CV) and natural language processing (NLP). The empirical results presented in this paper fill this gap and contribute significantly to the research of MRL. Specifically, the paper comprehensively explores how model performance changes across various dimensions, which is highly valuable to me.

2. The paper demonstrates a comprehensive consideration of settings for most dimensions and presents reasonable result analysis. When the observed trends align with those in other domains, it confirms the applicability of neural scaling laws across diverse fields. For dimensions where unexpected observations or unexplainable phenomena arise, intriguing future research directions are motivated.

3. Considering the promising applications of MRL in virtual screening and other aspects of drug design, this contribution has significant societal value, making it beneficial to society at large.

**Additional Feedback:**

None

**Clarity:**

This paper is well-written, where the core idea can be effortlessly picked. The notations are kept simple, and the figures and tables are informative and easy-to-understand.

**Correctness:**

This submission is mainly a benchmark. The methods seem to be correctly implemented, and the experiments are designed in a reasonable and practical way. Thus, I would like to trust the presented empirical results and make my research based on these findings.

**Documentation:**

To reproduce this benchmark, datasets involved are public, and the considered methods are popular and have been extensively compared. Besides, the experimental settings are detailed in this paper, which makes it viable to exactly reproduce the reported results.

**Opportunities For Improvement:**

1. As the authors have discussed, the considered neural architectures are too limited. Actually, Graph Transformer models have shown their advantages over traditional message-passing-based GNNs in many MRL tasks. It is necessary to study their behaviors and properties.
2. To motivate the future works in data pruning for MRL, it must be helpful to investigate the good/bad selections and report their uniqueness.
3. line44 mentions "graph", which is confusing to me, when I eventually find that this paper has compared both simple  planar graph and 3D point cloud.

**Relation To Prior Work:**

To my knowledge, this is the first work to study the neural scaling laws in MRL.

**Summary And Contributions:**

This paper investigates the neural scaling laws in molecular representation learning (MRL), which is currently one of the hottest topics in AI4Science. Considering the strong demand for such research and the lack thereof, the empirical results presented in this paper are expected to be beneficial for the AI4Science community. Specifically, this study examines various dimensions of MRL, revealing surprising trends in certain dimensions while observing consistency with other domains in others. Regardless of the observed trends, the empirical findings for each specific dimension imply valuable research questions that need to be addressed.

---

> ### Author Response · Authors · 2023-08-21
> **Response to Reviewer jova**
>
> We would like to express our sincere appreciation for your positive feedback on the significance of our work and the constructive suggestions provided. We carefully considered each comment and have addressed the concerns raised. Below, we provide a point-to-point response to the comments:
>
> - **As the authors have discussed, the considered neural architectures are too limited. Actually, Graph Transformer models have shown their advantages over traditional message-passing-based GNNs in many MRL tasks. It is necessary to study their behaviors and properties.**
>
> Thank you for your insightful feedback. During our preliminary experimental exploration, we did consider employing GT as a backbone model for investigating neural scaling laws. In particular, we utilized the GraphGPS [1] framework to evaluate its performance in molecular property prediction tasks with provided hyperparameters in their paper. The results are shown as below:
>
> |                       |      HIV      |      MUV      |     PCBA      |
> | :-------------------: | :-----------: | :-----------: | :-----------: |
> | **GraphGPS**-scaffold | 69.86$\pm$4.1 | 66.48$\pm$0.2 | 19.02$\pm$0.7 |
> |   **GIN**-scaffold    | 74.24$\pm$0.3 | 73.10$\pm$2.6 | 21.58$\pm$0.1 |
> |  **GraphGPS**-random  | 70.47$\pm$2.2 | 66.60$\pm$2.8 | 25.82$\pm$0.7 |
> |    **GIN**-random     | 86.21$\pm$2.1 | 80.46$\pm$0.3 | 28.19$\pm$0.1 |
>
> The results indicate that GraphGPS consistently underperforms GIN. Furthermore, during our experiments, we found that the performance of GraphGPS is highly sensitive to hyperparameter settings. In contrast, GNN models demonstrate robust performance across a wide range of datasets and experimental settings without complex hyperparameter tuning. Also, the GNN models have much fewer parameters, which effectively mitigates computational costs and aligns favorably with practical applications in drug discovery.
>
> Considering these factors, the current usage of GT model in the MRL domain remains nascent, rendering it insufficient to serve as a candidate backbone for our benchmark. We believe in the potential of GT models for MRL and will leave this exploration for future works.
>
> *[1] Recipe for a General, Powerful, Scalable Graph Transformer. NeurIPS 2022*
>
> - **To motivate the future works in data pruning for MRL, it must be helpful to investigate the good/bad selections and report their uniqueness.**
>
> We appreciate your constructive feedback. As mentioned in Section 3.6, the six data pruning methods we benchmarked are originally proposed for image data. Therefore, the reason why their performance did not surpass random selection is likely due to the discrepancy between image and molecular data. We believe that this topic demands a much deeper exploration that is out of the scope of our model-focused benchmark.
>
> - **line44 mentions "graph", which is confusing to me, when I eventually find that this paper has compared both simple planar graph and 3D point cloud.**
>
> Thank you for your meticulous review. The "graph" mentioned in line 44 refers to the 2D graph (planar graph), where nodes represent atoms and edges represent chemical bonds. In our revised manuscript, we distinguish planar graph and 3D point cloud as 2D graph and 3D graph to avoid potential confusion.

---

> > ### Comment · Reviewer_jova · 2023-08-23
> > **Discussion**
> >
> > Thanks for your response. Most of my concerns are resolved now. I also read the discussions with other reviewers. I will keep the score to support this paper.

---

> > > ### Author Response · Authors · 2023-08-23
> > > **Response to Reviewer jova**
> > >
> > > Thank you for your prompt feedback. I'm glad to hear that most of your concerns have been addressed. Your support is greatly appreciated. If you have any further questions or need additional information, please don't hesitate to reach out.

---

### Official Review · Reviewer_weCW · 2023-07-24
**Interesting work although some more discussion would help**

**Rating:** 6
**Confidence:** 3
**Correctness:** The claim in the work seem reasonable…

**Strengths:**

(S1): The work investigates important research questions, outlines them neatly, and shows experiments which are (for the most part) relatively convincing. The general findings are practical, and certainly useful for practitioners working with molecular models.

(S2): In particular, the analysis comparing the scaling behaviour of different modalities (graph, SMILES, fingerprint) is interesting and timely.

**Additional Feedback:**

Some extra nitpicks that came to my mind when reading the paper are listed below. I just highlight them in case they are helpful; feel free to just ignore those you disagree with.

- "Please relegate more details about (…)" - is this a normal/expected use of the word "relegate", or did you mean "delegate"?

- Missing dot at the end of Section 2.2

- "ASCII characters, with explicitly depicting information" - the "with" here sounds a bit odd to me, and I'm not sure if using "depict" for a text-based (non-visual) representation is normal

- "our empirical analysis cover" - "covers"?

- "shown as follow" - "follows"?

- Figure 4 says the outliers are red, yet they look gray to me

- Space before dot on line 259

- "modal capacity" - "model"?

- "direction which remain unexplored" - "remains"?

**Clarity:**

The paper is well-written and clear, but one issue I found is that the pdf file itself is slightly "broken" for me: I cannot highlight any text by dragging across the document, nor could I search in the text or follow any links/citations. This does not happen for any other paper, just this one; it looks as if the pdf contained pictures of text and not actual text. This made it hard to review, as I couldn't make notes on the pdf as normal. Assuming others are experiencing the same, I would strongly suggest the authors fix it.

**Documentation:**

The focus here is on empirical analysis and not introducing new tasks/datasets, hence some of these points do not apply. Nevertheless, there seem to be enough details to ensure reproducibility.

**Ethics:**

I see no ethical concerns.

**Limitations:**

Limitations and societal consequences are sufficiently discussed.

**Opportunities For Improvement:**

(W1): I think the performance degradation of the SMILES encoder is a rather weird phenomenon, and it would be good to have at least some hypothesis for why it happens. It's especially odd that there is no need to look at a very large range of dataset sizes for this to happen, i.e. it's observed going from 30% to 100%.

(W2): I think it would be nice to have a bit more discussion on the encoder choice for the fingerprint modality. If I understand correctly, the encoder is a transformer acting over the bit sequence dimension. Do you add standard positional encodings? Is the bit order fixed but essentially random (i.e. bits adjacent in the sequence have no particular relation to each other)? If this architecture was explored in the past then maybe add some references too. I'm curious if this works better than the usual MLP-based architecture (without a comparison, one could say that the MLP-on-fingerprints encoder should have been used as the more standard approach).

(W3): Figure 5 explores scaling of the data size with several choices of model size. How about scaling model size (over a wider range perhaps) with fixed large data size? Would that not be a good addition to the comparisons made in this work, and follow how scaling laws were commonly investigated in e.g. NLP?

**Relation To Prior Work:**

The paper is reasonably grounded in prior work, except perhaps for using a slightly unusual encoder on molecular fingerprints without providing references (see (W2) for details).

**Summary And Contributions:**

This paper outlines several research questions in molecular representation learning, for example touching on how varying data/model size and molecular modality/encoder impacts results across various tasks. The authors then perform a sequence of experiments to analyse the trends and answer each question in turn.

---

> ### Author Response · Authors · 2023-08-21
> **Response to Reviewer weCW (1/3)**
>
> - **I think the performance degradation of the SMILES encoder is a rather weird phenomenon, and it would be good to have at least some hypothesis for why it happens. It's especially odd that there is no need to look at a very large range of dataset sizes for this to happen, i.e. it's observed going from 30% to 100%.**
>
> We appreciate your detailed observations and suggestions. As mentioned in Section 3.2, the observed performance degradation of the SMILES encoder was confined to the MUV dataset. Given that MUV constitutes a multi-task challenge, we show in the appendix C.4 the neural scaling laws of single-property performance and offer relevant observations. Here, we hypothesize that the performance decline could be linked to the unique attributes of the MUV dataset itself.
>
> Previous research [1, 2] has shown that models explicitly embedding substructure information tend to excel in MUV performance. Interestingly, models employing the SMILES modality, such as Molformer [3] and ChemBERTa [4], often lack evaluations on the MUV dataset. Thus, we conjecture that the distinctive nature of MUV might be closely associated with molecular substructures or functional groups.
>
> Regrettably, apart from the above observation and conjecture, we are unable to provide a definitive scientific explanation for the performance decline of the SMILES encoder in terms of data utilization efficiency within the context of MUV. We believe this is beyond the scope of our benchmarking analysis and we shall leave this question for future exploration.
>
> [1] *Molecular Contrastive Learning of Representations via Graph Neural Networks. Nat. Mach. Intell* 2022.
>
> [2] *Improving Molecular Contrastive Learning via Faulty Negative Mitigation and Decomposed Fragment Contrast. J. Chem. Inf. Model 2022.*
>
> [3] *Large-scale chemical language representations capture molecular structure and properties. Nat. Mach. Intell 2022.*
>
> [4] *Large-Scale Self-Supervised Pretraining for Molecular Property Prediction. Arxiv 2020.*

---

> ### Author Response · Authors · 2023-08-21
> **Response to Reviewer weCW (2/3)**
>
> -  **I think it would be nice to have a bit more discussion on the encoder choice for the fingerprint modality. If I understand correctly, the encoder is a transformer acting over the bit sequence dimension. Do you add standard positional encodings? Is the bit order fixed but essentially random (i.e. bits adjacent in the sequence have no particular relation to each other)? If this architecture was explored in the past then maybe add some references too. I'm curious if this works better than the usual MLP-based architecture (without a comparison, one could say that the MLP-on- fingerprints encoder should have been used as the more standard approach).**
>
> Thank you very much for your valuable suggestion. Here is our response to your comments:
>
> **(1)** We did add standard positional encodings and have provided a detailed introduction to the transformer-based fingerprint encoder in Appendix A. Equations (7) and (8) therein illustrate our utilization of standard positional encodings.
>
> **(2)** Morgan fingerprints [5, 6] encode molecules as fixed-length binary strings, wherein bits denote the presence or absence of specific substructures. Each atom is represented based on a set of atomic invariants, and these features are iteratively updated across neighboring atoms using a hash function. However, it's important to note that the implementation of this hash function is deterministic. In other words, the same SMILES string will yield the same fingerprint. Therefore, it is reasonable to employ positional encoding to capture the sequential relationship of bits in the fingerprints.
>
> **(3)** Your concern about the choice of fingerprint encoder is valid. To the best of our knowledge, prior research has not explicitly reported results with FPs+MLP. However, we did explore using an MLP as the encoder, and below are our experimental results:
>
> |                  | 1%     | 5%     | 10%    | 20%    | 30%    | 40%    | 60%    | 80%    | 100%   |
> | ---------------- | ------ | ------ | ------ | ------ | ------ | ------ | ------ | ------ | ------ |
> | HIV-MLP          | 0.5590 | 0.6875 | 0.7384 | 0.7923 | 0.8109 | 0.8278 | 0.8317 | 0.8443 | 0.8360 |
> | HIV-Transformer  | 0.6311 | 0.7231 | 0.7524 | 0.7995 | 0.8118 | 0.8189 | 0.8325 | 0.8430 | 0.8487 |
> | MUV-MLP          | 0.5487 | 0.5762 | 0.5994 | 0.6166 | 0.6548 | 0.6458 | 0.6737 | 0.6976 | 0.7288 |
> | MUB-Transformer  | 0.5153 | 0.6069 | 0.5997 | 0.6526 | 0.6838 | 0.6957 | 0.7115 | 0.7230 | 0.7313 |
> | PCBA-MLP         | 0.0444 | 0.0776 | 0.1011 | 0.1279 | 0.1433 | 0.1539 | 0.1699 | 0.1760 | 0.1865 |
> | PCBA-Transformer | 0.0508 | 0.0846 | 0.1060 | 0.1315 | 0.1421 | 0.1542 | 0.1687 | 0.1775 | 0.1871 |
>
> Results indicate that MLP yields inferior performance compared to the transformer while still adhering to the power law relationship. We have added the corresponding explanations and empirical results to the revised main text and Appendix C.3. This will elucidate our rationale for selecting the transformer as the fingerprint backbone model.
>
> [5] *The Generation of a Unique Machine Description for Chemical Structures — A Technique Developed at Chemical Abstracts Service. J. Chem. Doc. 1965.*
>
> [6] *Circular Fingerprints: Flexible Molecular Descriptors with Applications from Physical Chemistry to ADME. IDrugs 2006.*
>
> - **Figure 5 explores scaling of the data size with several choices of model size. How about scaling model size (over a wider range perhaps) with fixed large data size? Would that not be a good addition to the comparisons made in this work, and follow how scaling laws were commonly investigated in e.g. NLP?**
>
> We greatly appreciate your constructive feedback. We believe the range we explored in this study, from [64$\times$2, 100$\times$3, 300$\times$5, 600$\times$10], is already enough. The reason we did not opt for a wider range of model sizes stems from certain limitations inherent to graph neural networks. According to the definition of model capacity, enlarging the model capacity could be done in two ways: increasing the network's depth or width. However, as depth increases, GNNs are prone to encountering issues such as over-smoothing [7] and over-squashing [8]. On the other hand, increasing the width by increasing the hidden dimensions could result in training instability.
>
> Therefore, the explored parameter range essentially spans the effective limits of viable model parameters. This coverage accounts for the constraints imposed by the aforementioned challenges and ensures a meaningful exploration within the context of GNNs.
>
> *[7] On the bottleneck of graph neural networks and its practical implications. ICLR 2021*
>
> *[8] Deeper Insights into Graph Convolutional Networks for Semi-Supervised Learning. AAAI 2018*

---

> ### Author Response · Authors · 2023-08-21
> **Response to Reviewer weCW (3/3)**
>
> - **The pdf file itself is slightly "broken"**
>
> We apologize for any inconvenience this may have caused. We have now provided a corrected PDF file that ensures proper functionality and is fully searchable.
>
> - **Some extra nitpicks that came to my mind when reading the paper are listed below.**
>
> We appreciate your meticulous review, and we have diligently revised the manuscript in accordance with your suggestions. Additionally, we have conducted a thorough proofreading to ensure that grammatical errors have been eliminated and have polished certain expressions for better readability.

---

> ### Author Response · Authors · 2023-08-24
> **Thank you & looking forward to reply**
>
> Dear Reviewer weCW:
>
> Thank you very much for your precious time and valuable comments. With the author-reviewer interaction period coming to a close in 5 days, we wanted to kindly remind you to take another look at our revisions if you haven't had a chance yet. We sincerely hope our responses have addressed all your concerns, but should there be any remaining issues or queries, please do not hesitate to let us know.
>
> Best,
>
> Authors

---

> > ### Comment · Reviewer_weCW · 2023-08-30
> > **Response**
> >
> > Thank you for your response! I think the paper is in good shape for publication, and all reviewers seem to agree on that. I maintain my positive rating. See below for some follow-up clarifications regarding some of the points discussed above.
> >
> > ---
> >
> > >  To the best of our knowledge, prior research has not explicitly reported results with FPs+MLP. However, we did explore using an MLP as the encoder, and below are our experimental results.
> >
> > Right, sorry, the most common baseline would be FPs + Random Forest, but FPs + MLP is also sometimes used; the former is especially common among practitioners (e.g. FP+RF is a strong baseline in "FS-Mol: A Few-Shot Learning Dataset of Molecules"). Thanks for providing the results though.
> >
> > > It's important to note that the implementation of [fingerprints] is deterministic. In other words, the same SMILES string will yield the same fingerprint. Therefore, it is reasonable to employ positional encoding to capture the sequential relationship of bits in the fingerprints.
> >
> > I know; what I meant is that bit `i` has no strong relationship with bit `i+1` despite them being next to each other, which is unlike e.g. NLP where subsequent words are strongly linked. The order of bits is therefore random, but always the same. Positional encodings have a bias towards treating "close by" tokens differently, which is not a useful bias here, but the network can just ignore this bias and use the encodings as identifiers of the bits. As you say, this makes sense as the `i`-th bit will always "mean the same thing" (i.e. same hash bucket of substructures, more or less) across all inputs.
> >
> > > We have now provided a corrected PDF file that ensures proper functionality and is fully searchable.
> >
> > Great, thanks!

---

### Official Review · Reviewer_5fTN · 2023-07-25
**Valuable topic but insufficient experiments**

**Rating:** 6
**Confidence:** 4
**Correctness:** Yes, the evaluation and experiment de…
**Clarity:** Yes, the paper is well written and ea…

**Strengths:**

1. The topic is valuable and underexplored. How deep learning models scale in molecular property predictions can provide helpful guidance in building larger and better molecular datasets.
2. The authors propose several interesting aspects of scaling law in MRL, including data quantity, modality, pre-training, etc.

**Additional Feedback:**

N/A

**Documentation:**

Yes, sufficient detail is provided.

**Ethics:**

No.

**Limitations:**

The authors have included discussions regarding the limitations of the works. For more potential limitations, please refer to "Opportunities For Improvement".

**Opportunities For Improvement:**

1. One of the major concerns is that even relatively large datasets (>40K) are selected. The scale is still much smaller than the datasets in vision and text. So the training cannot really "scale up" due to the limited data.
2. Following comment 1, the authors mention in Observation 1 that there is no obvious plateau like NLP and CV. Can it be because the training data is limited so that the models are showing similar performance as scaling in NLP and CV?
3. The authors use 4 modalities in section 3.1, but the remaining sections heavily focus on graphs only. This makes the observations incomplete and the conclusions cursory.
4. The authors use fixed architectures (e.g., 50% dropout in GIN). Can such settings limit the performance of models?
5. What's the multitask learning setting? The detail seems to be missing in the manuscript.
6. The investigation of the pre-training effect on scaling law is incomplete as the authors mention in the limitations. The authors only apply one pre-training method on graphs and draw a hasty conclusion that pre-training only benefits small downstream datasets. There are other pre-training strategies for different modalities that are neglected in this work.
7. In section 3.4, it seems that models trained under scaffold and imbalanced splitting don't well obey the scaling law. Are you suggesting that random splitting is a more realistic setting and should be applied in practice?
8. In line 44, only graphs, SMILES, and fingerprints are mentioned. However, 3D geometry is also included in the experiments and should be added.

**Relation To Prior Work:**

Yes.

**Summary And Contributions:**

In this work, the authors investigate the scaling law in molecular representation learning (MRL). The authors propose several aspects in this area, including data quantity, modality, pre-training, data distribution, model capacity, etc. To this end, GIN is implemented for 2D graphs, SchNet for 3D geometry, and Transformer for Morgan fingerprint/SMILES. The work mainly benchmarked on four relatively large datasets from MoleculeNet, namely HIV, MUV, PCBA, and QM9. Through experiments, the authors conclude that the scaling law applies to MRL in general. And modality plays an important role in MRL.

---

> ### Author Response · Authors · 2023-08-21
> **Response to Reviewer 5fTN (1/2)**
>
> - **One of the major concerns is that even relatively large datasets (>40K) are selected. The scale is still much smaller than the datasets in vision and text. So the training cannot really "scale up" due to the limited data.**
>
> Thank you for your constructive comments. In order to address your concern regarding the data scale, we have conducted additional scaling law experiments using the PCQM4Mv2, a large-scale challenge dataset provided by OGB ([https://ogb.stanford.edu/docs/lsc/pcqm4mv2](https://ogb.stanford.edu/docs/lsc/pcqm4mv2/)). To the best of our knowledge, it is currently the largest labeled molecular dataset available, containing over 3.7 million molecular samples. It is worth noting that due to the unavailability of official test data, we selected 50% from the validation set as the test data. Here are the results of our experiments:
>
> | **Summary** | **1%** | **5%** | **10%** | **20%** | **30%** | **40%** | **60%** | **80%** | **100%** |
> | ----------- | ------ | ------ | ------- | ------- | ------- | ------- | ------- | ------- | -------- |
> | 0           | 0.4229 | 0.3246 | 0.3004  | 0.2765  | 0.2676  | 0.2786  | 0.2660  | 0.2580  | 0.2546   |
> | 1           | 0.4190 | 0.3298 | 0.3022  | 0.2761  | 0.2724  | 0.2655  | 0.2621  | 0.2594  | 0.2537   |
> | 2           | 0.4240 | 0.3336 | 0.3062  | 0.2735  | 0.2773  | 0.2705  | 0.2655  | 0.2625  | 0.2650   |
> | avg         | 0.4220 | 0.3293 | 0.3029  | 0.2754  | 0.2724  | 0.2715  | 0.2645  | 0.2600  | 0.2578   |
> | std         | 0.0027 | 0.0045 | 0.0030  | 0.0017  | 0.0049  | 0.0066  | 0.0021  | 0.0023  | 0.0063   |
>
> As observed, our conclusions from the submitted manuscript remain consistent on the PCQM4Mv2 dataset, demonstrating a power-law relationship between model performance and data scale without clear signs of a plateau. We have incorporated these results in the revised figure 2. Please refer to Section 3.1 for detailed description.
>
> - **Following comment 1, the authors mention in Observation 1 that there is no obvious plateau like NLP and CV. Can it be because the training data is limited so that the models are showing similar performance as scaling in NLP and CV?**
>
> Thank you for your further comments about the data scale. As the results we presented in the first response, with an almost 10x larger dataset PCQM4Mv2, there is still no obvious plateau in the model performance.
>
> In our revised manuscript, we have modified this conclusion as: **there is no obvious plateau under the explored scale of our experiments.** We appreciate your insightful question, and we have included the aforementioned results and explanations in the revised manuscript.
>
> *[1] Scaling Laws for Deep Learning Based Image Reconstruction. ICLR 2023*
>
> - **The authors use 4 modalities in section 3.1, but the remaining sections heavily focus on graphs only. This makes the observations incomplete and the conclusions cursory.**
>
> To improve the coverage of this study, we have considered additional experiments to provide a more comprehensive analysis:
>
> 1. **Pre-training with SMILES and 3D Graph Modality**: While our main focus was on pre-training with the graph modality, we acknowledge the importance of exploring the effects of pre-training on SMILES and 3D graph modality. For fingerprint, it lacks existing literature on this direction, so we will leave the investigation of fingerprint pre-training effects as future work.
> 2. **Data Distribution with Fingerprint and SMILES Modality**: We have conducted experiments on the impact of data distribution with fingerprint and SMILES modality. As for 3D modality, the literature has not yet reached a consensus on splitting the data distribution. For detailed results, please refer to Appendix C.2 of our revised manuscript.
> 3. **Model Capacity with Fingerprint and SMILES**: We will further conduct experiments on the impact of model capacity with fingerprint and SMILES modality. In [2], the scaling law of model capacity for the 3D modality was explored and generally it also adheres to the power law relationship. Therefore, in light of the existing study, we will not explore the effect of model capacity with 3D modality in our paper.
>
> Given the time limitations for the rebuttal, we are actively working on experiments 1 and 3, but we find it difficult to finish all these experiments on time. We promise to include these results in the final version of the paper.
>
> *[2] Neural scaling of deep chemical models. Arxiv 2022.*

---

> ### Author Response · Authors · 2023-08-21
> **Response to Reviewer 5fTN (2/2)**
>
> - **The authors use fixed architectures (e.g., 50% dropout in GIN). Can such settings limit the performance of models?**
>
> Thank you for your question. As mentioned in our manuscript, the fixed architectures we utilized, with the exception of the fingerprint modality, are well-established and widely recognized model structures in the field. The settings for GIN are based on the configuration by Hu et al. [3], the settings for SchNet architecture follows Schütt et al.'s specifications [4], and the Transformer architecture is also commonly employed for SMILES data [5, 6].
>
> We investigate the corresponding scaling law relationships based on these effective and classical models. It is worth noting that the primary focus of our study is not on achieving the highest performance, but rather on understanding the scaling behavior within the context of these established architectures.
>
> *[3] Strategies for pre-training graph neural networks. ICLR 2020.*
>
> *[4] A Continuous-filter Convolutional Neural Network for Modeling Quantum Interactions. Arxiv 2017.*
>
> *[5] Large-Scale Self-Supervised Pretraining for Molecular Property Prediction. Arxiv 2020.*
>
> *[6] Large-scale chemical language representations capture molecular structure and properties. Nat. Mach. Intell 2022.*
>
> - **What's the multitask learning setting? The detail seems to be missing in the manuscript.**
>
> We apologize for any unclarity in this regard. In fact, the multitask learning setting refers specifically to the MUV and PCBA datasets. As indicated in Table 2 in the appendix, MUV and PCBA dataset involves 17 and 92 tasks, respectively. This signifies that these datasets simultaneously make predictions for multiple properties, rather than being multi-class classification tasks. We have addressed your concern by adding explanations to the revised manuscript.
>
> - **The investigation of the pre-training effect on scaling law is incomplete as the authors mention in the limitations. The authors only apply one pre-training method on graphs and draw a hasty conclusion that pre-training only benefits small downstream datasets. There are other pre-training strategies for different modalities that are neglected in this work.**
>
> Thank you for your constructive comments. As mentioned in our previous response, we are actively working on conducting the investigation of pre-training effects across different modalities and will report the results once the experiments are finished.
>
> - **In section 3.4, it seems that models trained under scaffold and imbalanced splitting don't well obey the scaling law. Are you suggesting that random splitting is a more realistic setting and should be applied in practice?**
>
> Thank you for your question. We would like to claim that based on the experimental results in section 3.4, only the MUV dataset under imbalanced split seems to deviate from the power law. However, all splitting methods follow the power law in all other datasets for the 2D graph modality. In any case, we could not make decisive conclusions about which splits to use in reality solely based on the scaling behavior of models. Our aim is to present the observed experimental phenomena under different splits and to highlight that random splitting is the most efficient way to leverage the training data. We emphasize that it does not mean it is a more realistic setting. This observation also motivates future work to study a more efficient way to handle OOD scenarios from a data-centric perspective.
>
> - **In line 44, only graphs, SMILES, and fingerprints are mentioned. However, 3D geometry is also included in the experiments and should be added.**
>
> Thank you for your careful review. In the context of Line 44, the section that follows (3.3) does not actually include a comparison involving the 3D modality. Our exploration of the scaling behavior of the 3D modality in the QM9 dataset was conducted in Section 3.1. Therefore, in that particular line, we only mentioned the three modalities under consideration. We have modified the wording in the revised manuscript and hope this clarification addresses your concern.

---

> ### Author Response · Authors · 2023-08-24
> **Thank you & looking forward to reply**
>
> Dear Reviewer 5fTN:
>
> Thank you very much for your precious time and valuable comments. With the author-reviewer interaction period coming to a close in 5 days, we wanted to kindly remind you to take another look at our revisions if you haven't had a chance yet. We sincerely hope our responses have addressed all your concerns, but should there be any remaining issues or queries, please do not hesitate to let us know.
>
> Best,
>
> Authors

---

> > ### Comment · Reviewer_5fTN · 2023-08-26
> > **Response to rebuttal**
> >
> > I thank the authors for answering my questions. I believe the extra experiments have greatly supplemented the work, especially:
> > 1. added equivariant GNNs for 3D molecules geometries besides invariant GNNs;
> > 2. benchmarked on much larger dataset PCQM4Mv2 and still demonstrated scaling law;
> > 3. more coverage of different modalities to be added in final version.
> > 4. added validation of the design choice of GNN architectures.
> >
> > Therefore, I have increased my score to 6.

---

> > > ### Author Response · Authors · 2023-08-26
> > > **Further response**
> > >
> > > We sincerely appreciate your feedback and the time you've dedicated to reviewing our work. Your comments have greatly improved our manuscript further.

---

### Official Review · Reviewer_rEJP · 2023-07-26
**A data-centric analysis of molecular representation learning dynamics**

**Rating:** 6
**Confidence:** 3

**Strengths:**

* The authors investigate a variety of molecular data modalities, highlighting the strengths (and weaknesses) of each.
* The authors perform their experiments using a variety of molecular datasets of various sizes and compositions.
* The authors succinctly highlight for readers the main takeaways of this work, section-by-section, allowing readers to closely follow the authors' line of reasoning behind their successive experiments.
* The authors provide well-documented source code on GitHub to rerun their experiments.

**Additional Feedback:**

This reviewer thanks the authors for raising awareness of the need for data-centric investigation of molecular representation learning dynamics.

**Clarity:**

* The paper is written fairly well, albeit with occasional typos and incomplete sentences. The authors should work to ensure their manuscript is free of such grammatical issues in future versions of the manuscript, to improve the readability of this work.

**Correctness:**

* The authors must cite a scientific source to ground their argument for using random splits as opposed to scaffold-based splits.

**Documentation:**

* The authors' provided GitHub is fairly well documented. However, it is advised that the authors package all their required software dependencies as a Conda/pip environment to make the work even more reproducible.

**Ethics:**

* No ethical concerns are raised by this work, insofar as the datasets investigated in this work are ethically constructed and deployed.

**Limitations:**

* The authors only investigate invariant graph representation learning methods for their study's proposed graph modality. Equivariant graph representation learning methods have recently been shown to be theoretically and empirically more powerful methods for encoding 3D molecular graph inputs. As such, the effects of neural molecular scaling laws on equivariant representations remain uninvestigated.

**Opportunities For Improvement:**

* The authors only provide regression results on the QM9 dataset in their first experiment. Moreover, the authors should clearly label the plots in Figure 1 belonging to QM9 tasks.
* The authors do not compare or discuss their findings in light of the work of Nathan Frey et al. "Neural scaling of deep chemical models." (2022). Such work, albeit focused on neural chemical models, seems to address related topics and discussion points relevant to this manuscript under review.
* There are some areas in the manuscript that appear to be incomplete. For example, Line 106 seems to be a note to the coauthors of this work.
* The manuscript PDF itself seems to not be searchable, possibly due to how it was exported. It is recommended to make the PDF searchable to help authors navigate this work.

**Relation To Prior Work:**

* The authors must discuss the relation of their work to Frey et al. 2022 (as mentioned in "Opportunities for Improvement" above). Besides this, the related works section seems a bit sparse regarding a discussion of different graph representation learning methods available today (e.g., SphereNet, SpookyNet, PaiNN, etc.).

**Summary And Contributions:**

The authors insightfully point out the need for data-centric investigations into the behavior of molecular representation learning methods. They provide a comprehensive look at (1) the existence of neural scaling laws for molecular data; (2) the impact of popular data modalities for molecules; (3) the effect of pre-training on the neural scaling laws discovered in (1); (4) the influence of data distribution on such neural scaling laws; (5) the way in which a model's capacity impacts the neural scaling laws; and (6) the overall harmful effects of standard (e.g., computer vision) data pruning strategies on molecular representation learning and scaling laws. The above discoveries and discussions provide useful insights for future research into molecular representation learning. Nonetheless, as outlined below, key points must be addressed before this work can be published.

---

> ### Author Response · Authors · 2023-08-21
> **Response to Reviewer rEJP (1/2)**
>
> - **The authors only provide regression results on the QM9 dataset in their first experiment. Moreover, the authors should clearly label the plots in Figure 1 belonging to QM9 tasks.**
>
> Thank you for your constructive comments and suggestions. In the revised manuscript, we have clearly labeled the regression results belonging to the QM9 dataset in Figure 1.
>
> Regarding the reason we only provide regression results in the first experiment, we make a point-to-point response as below:
>
> 1. **Modality**: The QM9 dataset provides quantum property prediction tasks, wherein 3D conformations are derived from Density Functional Theory (DFT) calculations. This direct relevance to quantum property prediction necessitates the utilization of the 3D modality for the QM9 dataset, as it aligns with established practices in the field. However, for datasets like MUV, HIV, and SMILES, the choice of modality is an open question. Thus, we put more attention at figuring out how these three modalities differ in scaling laws.
> 2. **Data split**: Due to the inability to extract scaffolds from some molecules in the QM9 dataset, we were unable to compare the performance of random split with scaffold split. As a result, QM9 was not included in the analysis of the impact of data split.
> 3. **Model capacity:** The paper "Neural scaling of deep chemical models" by Frey et al. has already extensively explored the scaling law for quantum property prediction tasks (in Figure A.3). In the revised manuscript, we will reference this work and discuss its relationship to our study. Given the existing literature, we do not conduct similar experiments in our work.
>
> 4. **The role of pre-training**: We apologize for overlooking the scaling law analysis related to 3D graph pre-training. Due to the long training time required for the QM9 dataset, we are still working on conducting the necessary experiments. We will provide relevant results once the experiments are finished to address your concern.
>
> - **The authors do not compare or discuss their findings in light of the work of Nathan Frey et al. "Neural scaling of deep chemical models." (2022). Such work, albeit focused on neural chemical models, seems to address related topics and discussion points relevant to this manuscript under review.**
>
> Thank you for bringing this article to our attention. Compared to "Neural scaling of deep chemical models" by Frey et al, we share a common perspective in analyzing Molecular Representation Learning (MRL) from a data-centric viewpoint. However, our research differs from this work in the following aspects:
>
> 1. **Scope of Neural Scaling Law Investigation:** The study of Frey et al. primarily considers large language models for *generative chemical modeling* and *quantum property* tasks. In contrast, our work encompasses a broader range of dimensions, such as modalities, data distributions, pre-training and model capacity, to investigate the influence of various factors on neural scaling laws.
> 2. **Different purposes:** Their contribution primarily revolves around developing strategies for scaling deep chemical models, notably large language models, and introducing novel model architectures. In contrast, our focus lies in exploring the impact of different dimensions on *data utilization efficiency* through a data-centric perspective and benchmarking existing data pruning strategies on MRL tasks.
>
> While sharing a common spirit, our research provides distinct insights that address a wide spectrum of factors affecting neural scaling laws in MRL.
>
> - **There are some areas in the manuscript that appear to be incomplete. For example, Line 106 seems to be a note to the coauthors of this work.**
>
> We apologize for any confusion caused by our wording. In Line 106, our intention was to direct readers to the relevant sections in the appendix for details regarding the datasets and MRL tasks. We have now revised the text in the manuscript.
>
> - **The manuscript PDF itself seems to not be searchable, possibly due to how it was exported. It is recommended to make the PDF searchable to help authors navigate this work.**
>
> We apologize for any inconvenience this may have caused. We have now provided a corrected PDF file and ensure it is fully searchable.

---

> ### Author Response · Authors · 2023-08-21
> **Response to Reviewer rEJP (2/2)**
>
> - **The authors only investigate invariant graph representation learning methods for their study's proposed graph modality. Equivariant graph representation learning methods have recently been shown to be theoretically and empirically more powerful methods for encoding 3D molecular graph inputs. As such, the effects of neural molecular scaling laws on equivariant representations remain uninvestigated.**
>
> We sincerely appreciate your insightful suggestion. Exploring the neural scaling laws for equivariant representations, specifically regarding the proposed 3D graph modality, is indeed a promising avenue. In response to this, we have extended our investigation to include SE(3) invariant model (SphereNet) and E(3) equivariant model (PaiNN) in the context of Section 3.1. Below, we present our experimental findings:
>
> | Summary        | 1%     | 5%     | 10%    | 20%    | 30%    | 40%    | 60%    | 80%    | 100%   |
> | -------------- | ------ | ------ | ------ | ------ | ------ | ------ | ------ | ------ | ------ |
> | PaiNN-U0       | 2.6910 | 0.5885 | 0.3760 | 0.2534 | 0.2036 | 0.1819 | 0.1318 | 0.1017 | 0.0939 |
> | PaiNN-gap      | 0.3537 | 0.1798 | 0.1246 | 0.0860 | 0.0703 | 0.0599 | 0.0480 | 0.0424 | 0.0396 |
> | PaiNN-zpve     | 0.0564 | 0.0135 | 0.0088 | 0.0052 | 0.0034 | 0.0025 | 0.0022 | 0.0019 | 0.0017 |
> | SphereNet-U0   | 2.3789 | 1.1014 | 0.7272 | 0.4383 | 0.4217 | 0.2749 | 0.1671 | 0.1496 | 0.1458 |
> | SphereNet-gap  | 0.3562 | 0.2094 | 0.1436 | 0.0856 | 0.0725 | 0.0636 | 0.0492 | 0.0472 | 0.0475 |
> | SphereNet-zpve | 0.0827 | 0.0756 | 0.0664 | 0.0362 | 0.0080 | 0.0043 | 0.0032 | 0.0021 | 0.0017 |
>
> Results indicate that all the performances generally adhere to the power-law relationship and the difference lies in the coefficients and exponents of power law. Detailed visualizations and relevant analyses are provided in Appendix C.1. We refer you to the supplementary experiments in the appendix for our additional observations in this direction.
>
> - **The authors must cite a scientific source to ground their argument for using random splits as opposed to scaffold based splits.**
>
> Thank you for your suggestion. As discussed in Section 3.4, our choice of data distribution does not disrupt the power-law relationship between model performance and data volume. This implies that our subsequent conclusions regarding neural scaling laws across various dimensions remain robust to changes in data distribution.
>
> Furthermore, the use of random splits is a more prevalent and commonplace experimental design in the realm of deep learning, as highlighted in our analysis. As an initial attempt in exploring multi-dimensional neural scaling laws within MRL, we aimed to adopt an experiment design that is generally applicable and avoids introducing potential confusion arising from the combined effects of scaffold and other dimension-related factors. The use of random splits ensures that the exploration of other factors remains unaffected by the influence of data distribution.
>
> - **The paper is written fairly well, albeit with occasional typos and incomplete sentences. The authors should work to ensure their manuscript is free of such grammatical issues in future versions of the manuscript, to improve the readability of this work.**
>
> We appreciate your meticulous review, and we have revised the manuscript in accordance with your suggestions. Additionally, we have conducted a thorough proofreading to eliminate grammatical errors and have polished certain expressions for better readability.
>
> - **The authors must discuss the relation of their work to Frey et al. 2022 (as mentioned in "Opportunities for Improvement" above). Besides this, the related works section seems a bit sparse regarding a discussion of different graph representation learning methods available today (e.g., SphereNet, SpookyNet, PaiNN, etc.)**
>
> Thank you for your constructive feedback. Regarding the comparison and discussion of the work by Frey et al., please refer to our previous responses. We have incorporated additional references and discussions in the revised manuscript.
>
> Furthermore, we have provided a more comprehensive review of molecular representation learning methods, particularly those related to 3D graph representation learning, in the related works section of the appendix. Please refer to Section D.1 in the appendix for an overview on this topic.
>
> - **The authors' provided GitHub is fairly well documented. However, it is advised that the authors package all their required software dependencies as a Conda/pip environment to make the work even more reproducible.**
>
> Thanks for your suggestions. We have incorporated an installation guidance in the GitHub repository for better reproducibility.

---

> ### Author Response · Authors · 2023-08-24
> **Thank you & looking forward to reply**
>
> Dear Reviewer rEJP:
>
> Thank you very much for your precious time and valuable comments. With the author-reviewer interaction period coming to a close in 5 days, we wanted to kindly remind you to take another look at our revisions if you haven't had a chance yet. We sincerely hope our responses have addressed all your concerns, but should there be any remaining issues or queries, please do not hesitate to let us know.
>
> Best,
>
> Authors

---

> > ### Comment · Reviewer_rEJP · 2023-08-24
> > **Response to Rebuttal by Authors**
> >
> > I would like to thank the authors for addressing several of my original concerns with this manuscript. I have since raised my score to a `6` in response. Below are some thoughts based on the authors' most recent rebuttals.
> >
> > (1) It is interesting to see that invariant and equivariant graph neural networks such as SphereNet and PaiNN both demonstrate power-law behavior like all other models.
> >
> > (2) The authors' expanded explanation of their choice of random splits is appreciated.
> >
> > (3) The authors' discussion of how their work is different from that of Frey et al. 2022 would be a good discussion point to include in the manuscript's supplementary material (or in the main text if there is room).
> >
> > (4) Have the authors also considered investigating scaling laws for larger molecular datasets such as GEOM-Drugs, since it also is well-posed as a 3D molecular graph dataset? This would determine whether or not neural scaling laws are closely affected by the size of a dataset's molecules. This certainly is not necessary for the purposes of this study, but in a follow-up study it would be interesting to see if similar scaling law trends hold for increasingly larger types of molecules such as proteins.
> >
> > (5) I would like to thank the authors for fixing the various formatting issues in the original manuscript's PDF file.
> >
> > (6) An important note about the GitHub repository. Even after the authors have added instructions for how to reproduce the authors' Python/Conda environment for software dependencies, it still is not clear how users can extend or modify the benchmarking code the authors have developed in this GitHub repository. Since there is no documentation (e.g., Sphinx webpages) or Jupyter notebook examples for how to build on top of this repository, it remains unclear how users will be able to leverage this code for future work.

---

> > > ### Author Response · Authors · 2023-08-24
> > > **Further response**
> > >
> > > We are pleased to hear that some of your original concerns have been addressed! Regarding your further feedback, please find our responses as follows:
> > > - Regarding the point (3), we have added a discussion comparing our work with that of Frey et al. into the Related Work section of the Appendix in our last revision. For further details, please refer to Appendix D.2.
> > > - Regarding the point (4), we acknowledge your suggestion and intend to extend our study to investigate the neural scaling law for GEOM-Drugs as part of the future work.
> > > - Regarding the point (6), we appreciate your suggestion and have added a Jupyter notebook example (neural scaling law of 2D graph modality with random split) to our GitHub repository to help users better leverage our code.

---

> > > > ### Comment · Reviewer_rEJP · 2023-08-25
> > > > **Comment on further response from the authors**
> > > >
> > > > (3) Thanks to the authors for adding an expanded discussion regarding this work and that of Frey et al.
> > > >
> > > > (6) Regarding the Jupyter notebook example, the code itself seems sufficient for beginners. However, one concern that still comes to mind is how easy it will be for new users of the authors' codebase to build on top of the authors' datasets and modeling code. I think it would be beneficial for the authors to create a standardized webpage of documentation for the authors' GitHub repository, e.g., using Sphinx. This could include lists of dataset classes, model classes, and utility functions new users may find useful or worth noting, along with simple docstrings for each. One can then host such documentation for free using GitHub Pages. This would greatly improve the usability of the existing codebase and reduce friction new users might experience when they set out to extend this codebase for future tasks. This is something that could be added to accompany this GitHub repository hopefully before e.g., a camera-ready deadline.

---

> > > > > ### Author Response · Authors · 2023-08-27
> > > > > **Response to Reviewer's comments about documentation**
> > > > >
> > > > > We appreciate your feedback regarding the usability of our codebase for new users. We have taken your recommendation into consideration and have created a standardized webpage of documentation MolScaling (https://molscaling.readthedocs.io/) for our GitHub repository using Sphinx. We invite you to take a look at it. Once again, thank you for your constrcutive comments, which further improves our work.

---

> > > > > > ### Comment · Reviewer_rEJP · 2023-08-29
> > > > > > **Thanks for including more documentation**
> > > > > >
> > > > > > I would like to thank the authors for including Sphinx webpages to accompany their manuscript and code repository. I would encourage the authors to continue to organize and expand their Sphinx documentation to make it even easier for its readers to acquire a depth and breadth of knowledge on the topic of scaling molecular deep learning models. For example, including more examples would be helpful, to show readers how to work with 3D modalities as well.
> > > > > >
> > > > > > Once again, thanks to the authors for their engagement during the rebuttal process.

---

> > > > > > > ### Author Response · Authors · 2023-08-30
> > > > > > > **Thanks for Reviewer's feedback**
> > > > > > >
> > > > > > > Thank you for your feedback and suggestions. We will keep expanding and updating our Sphinx documentation in the future, including more examples, to enhance the usability of our codebase. We appreciate your support and engagement during the rebuttal process.

---

### Official Review · Reviewer_pZ6H · 2023-07-27
**Review of paper "Uncovering Neural Scaling Law in Molecular Representation Learning"**

**Rating:** 6
**Confidence:** 2
**Clarity:** Yes

**Strengths:**

- Interesting empirical study investigating the performance of Molecular Representation Learning
- Extensive experiments and detailed results, well documented
- Interesting to see empirical evidence that MRL performace exhibits a power-law relationship with data quantity

**Additional Feedback:**

No additional feedback.

**Correctness:**

- The claims seems to be correct, and the study seems technically sound.

**Documentation:**

- I think authors could provide more information on the structure of the processed datasets used for their experiments (Google Drive link https://drive.google.com/drive/folders/1sWrG8ZhBvx9lrfzMHEhbEpLPjHuBdjm_?usp=drive_link)

**Limitations:**

- Authors may provide some additional information on the limitations of their study.
- I do not see any  negative societal impact of this work.

**Opportunities For Improvement:**

- Describe how your results may influence and/or affect future research in the area of  Molecular Representation Learning: is there any specific insights from your study that researchers in MRL should take into account. Summarize these insights.

**Relation To Prior Work:**

Good

**Summary And Contributions:**

The paper presents an empirical study on the  neural scaling behaviors of Molecular Representation Learning (MRL)  across various dimensions, including (1) data modality, (2) data distribution, (3) pre-training intervention, and (4) model capacity. The study confirms that the performance of MRL exhibits a power-law relationship with data quantity across four dimensions

---

> ### Author Response · Authors · 2023-08-21
> **Response to Reviewer pZ6H**
>
> - **Describe how your results may influence and/or affect future research in the area of Molecular Representation Learning: is there any specific insights from your study that researchers in MRL should take into account. Summarize these insights.**
>
> We thank you for your comments. Here, we provide a summary of our insights:
>
> Our empirical analysis across several dimensions reveals that the performance of MRL adheres to the power law relationship as the data quantity varies, implying the marginal effect of adding more data. Changes within different dimensions often manifest in varying data utilization efficiencies, characterized by alterations in power exponents and coefficients.
>
> In our revised manuscript, we have incorporated the suggested elucidations and highlight the modified content in the Section 4.
>
> - **Authors may provide some additional information on the limitations of their study.**
>
> Thank you for your thorough review and insightful feedback. We have included additional discussions regarding the limitations of our study as follows:
>
> 1. While we have endeavored to explore the relationships between various dimensions and data utilization efficiencies, our exploration has concentrated solely on predictive tasks, leaving an unexplored domain of generative tasks.
> 2. Even though the scaling law applies to most of the experiments, we do observe some violations worth further investigation.
>
> Please refer to Section 4 for detailed revision.
>
> - **I think authors could provide more information on the structure of the processed datasets used for their experiments (Google Drive link)**
>
> We greatly appreciate your valuable suggestions. In response to this, we have enriched the associated Google Drive link and README.md file with pertinent descriptions.

---

> ### Author Response · Authors · 2023-08-24
> **Thank you & looking forward to reply**
>
> Dear Reviewer pZ6H:
>
> Thank you very much for your precious time and valuable comments. With the author-reviewer interaction period coming to a close in 5 days, we wanted to kindly remind you to take another look at our revisions if you haven't had a chance yet. We sincerely hope our responses have addressed all your concerns, but should there be any remaining issues or queries, please do not hesitate to let us know.
>
> Best,
>
> Authors

---

> > ### Comment · Reviewer_pZ6H · 2023-08-25
> > **Response to authors comment and revised manuscript**
> >
> > I want to thank the authors to address my comments.

---

### Author Response · Authors · 2023-08-21
**General Response**

We would like to thank all reviewers very much for their extensive reviews and constructive critiques. We are encouraged that reviewers find that our experiments are extensive and results are detailed. (Reviewer PZ6H), that explored molecular datasets are of various sizes and compositions. (Reviewer rEJP), that the topic is valuable and investigations are important research questions (Reviewer 5fTN and weCW), and that our contribution has significant societal value (Reviewers jova). In response to your critiques, we have made several important additions and modifications (marked as red in the revised PDF), as outlined below:

**New Results**

1. We have included results using an MLP encoder for Fingerprint tasks, providing a comprehensive comparison with Transformer architecture in our experiments.
2. Experimental results for Spherenet and PaiNN models have been added, enhancing the diversity of 3D graph representation results.
3. To validate our findings on a larger scale, we have conducted experiments on the PCQM4Mv2 dataset and discussed the results.
4. Additional results have been provided for different data splits, focusing on Fingerprint and SMILES modalities.

**Modifications**

1. We have revised the text to provide a more detailed explanation of the insights generated from our study.
2. Further elaboration has been added to explain multi-task learning, addressing the specific concern.
3. A more comprehensive explanation of the limitations of our study has been provided.
4. Additional reference and discussions regarding deep chemical models have been incorporated into the manuscript.
5. Our review of molecular representation learning in the related works section has been expanded for better context.
6. Detailed information about the structure of the processed datasets used in our experiments is available through the provided Google Drive link.

We hope that these additions and modifications address your concerns and enhance the quality and comprehensiveness of our manuscript. We are grateful for your thoughtful evaluation and look forward to further discussions.

---

### Decision · Program_Chairs · 2023-09-22

**Decision:**

Accept (Poster)

**Comment:**

## Summary

The paper computes empirical scaling laws for molecular property prediction with a focus on how data scale and distribution affects test metrics.

## Strengths

- [quality, clarity, significance] The work investigates important research questions, outlines them neatly, and shows convincing experiments. The general findings are practical, and certainly useful for practitioners working with molecular models.
- [significance, originality] The topic is valuable and underexplored. How deep learning models scale in molecular property predictions can provide helpful guidance in building larger and better molecular datasets.
- [quality] The paper considers a variety of prediction tasks and models.

## Weaknesses

- Compared to the datasets used to compute scaling laws in CV and NLP, all the datasets used here are quite small.
- Due to mostly focusing on GNNs, the paper is unable to compute scaling laws on model capacity over a wider range of capacities.
- The pretraining section only considers one pre-training method, weakening those conclusions.

## Explanation of recommendation

Although most reviewers gave the paper a 6, I do not see any major unaddressed concerns. On my reading, while the paper could make broader statements with bigger datasets or models, the work presented is well-done and has interesting conclusions for the community.